

# Are atmospheric PBDE levels declining in Central Europe? Examination of the seasonal variations, gas-particle partitioning and implications for long-range atmospheric transport

Céline Degrendele[1], Jake Wilson[2], Petr Kukučka[1], Jana Klánová[1] and Gerhard Lammel[1,2]

[1]Masaryk University, Research Centre for Toxic Compounds in the Environment, Kamenice 5, 625 00 Brno, Czech Republic
[2]Max Planck Institute for Chemistry, Multiphase Chemistry Department, Hahn-Meitner-Weg 1, 55128 Mainz, Germany

*Correspondence to*: Céline Degrendele (degrendele@recetox.muni.cz)

**Abstract.** This study presents novel multi-year monitoring data on atmospheric polybrominated diphenyl ethers (PBDEs) in Central Europe. Air was sampled on a weekly basis at a background site in the central Czech Republic from 2011 to 2014 (N=114). $\Sigma_9$PBDEs (without BDE209) total (gas and particulate) concentrations ranged from 0.088 to 6.08 pg m$^{-3}$, while BDE209 was at 0.05-5.01 pg m$^{-3}$. BDE47, BDE99 and BDE183 were the major contributors to $\Sigma_9$PBDEs.

The inverse of ambient temperature was positively correlated with the total concentrations of BDE47 and BDE66 while negative correlations were observed for BDE153 and BDE154. Overall, the atmospheric concentrations of individual PBDEs were controlled by primary emissions, re-volatilisation from surfaces, deposition processes and long-range atmospheric transport.

Regarding gas-particle partitioning, with the exception of BDE28 (gaseous) and BDE209 (particulate), all congeners were consistently detected in both phases and clear seasonal variations were observed. For example, while the average measured particulate fraction ($\theta_{measured}$) of BDE47 was 0.52 in winter, this was only 0.01 in summer. Similarly, for BDE99, $\theta_{measured}$ was 0.88 in winter, while it was only 0.18 in summer. The observed gas-particle partitioning coefficient ($K_p$, in m$^3$ µg$^{-1}$) was compared with three model predictions, assuming equilibrium or steady-state. None of the models could provide a satisfactory prediction of the partitioning, suggesting the need for a universally applicable model.

Statistically significant decreases of the atmospheric concentrations during 2011-2014 were found for BDE100, 99, 153 and 209. Estimated apparent atmospheric halving times for these congeners were ranging from 2.6 (BDE209) to 4.5 (BDE153) years. Significant increases of BDE28 and 66 were observed for autumn and summer, respectively. This may suggest the accumulation of these persistent products in the environment due to photolytic conversion of higher brominated congeners.

## 1 Introduction

Since the late 1960s, flame retardants such as polybrominated diphenyl ethers (PBDEs) have been used in large quantities in various consumer products such as plastics, textiles, electronics and cars (Besis and Samara, 2012). Three main commercial formulations were produced: Penta-BDE, Octa-BDE and Deca-BDE. The major congeners in each formulation were BDE99





and BDE47 for the Penta mixture, BDE183 for the Octa mixture and BDE209 for the Deca mixture (La Guardia et al., 2006). The Deca mixture has been the most widely used, accounting approximately for 83% of the total PBDEs production worldwide (Besis and Samara, 2012). PBDEs are widespread contaminants as they are persistent, bioaccumulative, toxic and prone to long-range atmospheric transport (LRAT). Therefore, all PBDE technical mixtures use and marketing were banned in the

European Union by 2008 (Besis and Samara, 2012), and have been included in the Stockholm Convention on Persistent Organic Pollutants (POPs) (UNEP, 2009).

Similar to other semi-volatile organic compounds (SOCs), once PBDEs enter the air, they are partitioned between the gaseous and the particulate phase. This partitioning is controlled by the physico-chemical properties of PBDEs, the meteorological parameters (i.e. temperature and relative humidity) and the abundance and composition of suspended particulate matter

(Lohmann and Lammel, 2004; Pankow, 1987). This partitioning will significantly affect their removal pathways (i.e. wet and dry deposition, photolysis, reaction with OH radicals) which are different for gases and particles (Wania et al., 1998) and therefore their mobility and their potential for LRAT (Bidleman et al., 1986). Knowledge about this partitioning is deficient, but is crucial to predict the environmental fate of PBDEs. Due to their physico-chemical properties, it is expected that lower brominated congeners such as BDE-28 are mainly present in the gaseous phase while higher brominated congeners such as

BDE-209 are >99% present in the particulate phase (Harner and Shoeib, 2002; ter Schure et al., 2004). However, results from individual studies at a global scale are contradictory. For example, some studies have found that the particulate fraction of PBDEs was small for most PBDEs investigated (e.g. <20%, Iacovidou et al., 2009). But other studies reported that the particulate fraction significantly increased with increasing degree of bromination for the same temperature (Davie-Martin et al., 2016; Möller et al., 2011; Strandberg et al., 2001; Su et al., 2009). Recently, Li and co-workers developed a new gas-

particle partitioning theoretical model for PBDEs based on the assumption that the equilibrium between both phases is not reached due to disturbances caused by wet and dry deposition (Li et al., 2015), but the universal applicability of this model still remains to be shown (Besis et al., 2017).

About a decade after the European ban on PBDEs, it is still unclear whether the concentrations in the global atmosphere are significantly declining or not due to the limited amount of ambient air monitoring data, particularly in Central Europe. In order

to understand whether primary or secondary sources are controlling the atmospheric concentrations of PBDEs, and hence to guide future control strategies, more data are needed to fill this gap.

The aim of this study is to provide novel multi-year monitoring data on atmospheric PBDEs at a background site in Central Europe and to assess whether the PBDEs atmospheric concentrations are significantly decreasing or not in the time span of four years. In particular, the seasonal and semi-long-term variations as well as the gas-particle partitioning of PBDEs were

investigated.



## 2 Methodology

### 2.1 Air sampling

Air was sampled at the Košetice observatory (49°34'24''N, 15°04'49''E), which is an established background site of the European Monitoring and Evaluation Programme (EMEP) network (Holoubek et al., 2007). The site is located in an

agricultural region in central Czech Republic. From January 2011 to December 2014, a high-volume air sampler (Digitel DH77 with $PM_{10}$ pre-separator) was used to collect weekly air samples. The sample volume was 5264 $m^3$ on average ($\approx$ 31.3 $m^3$ $h^{-1}$, 7-day sampling duration). Particles were collected on quartz fiber filters (QFFs, QM-A, 150 mm, Whatman, UK, pore size of 2.2 µm) and gas-phase on polyurethane foam (PUF, two in series, T3037, 110 x 50 mm, 0.030 g $cm^{-3}$, Molitan a.s., Czech Republic). PUFs were pre-cleaned via Soxhlet extraction with acetone and dichloromethane for 8 h each. PBDEs analysis was

performed on all weekly samples in 2011 and on half of the available weekly samples for the remaining years (Table S1 in the Supplement). Several problems (e.g. sudden change in the flow rate, electrical power shutdown) occurred during the collection and the corresponding 13 samples were discarded for further analysis (Table S1). After sampling, all filters and PUFs were wrapped in aluminium foil, sealed in plastic bags and stored at -18°C until analysis.

### 2.2 Sample preparation and analysis

Samples were extracted with dichloromethane by means of an automated extraction system (Büchi B-811, Switzerland). Mass-labelled internal standards ($^{13}$C labelled BDE-28, BDE-47, BDE-99, BDE-100, BDE-153, BDE-154, BDE-183 and BDE-209, Wellington Laboratories, Canada, LGC, UK) were added to each sample prior to extraction. The clean-up and fractionation method were adopted from a method used for polychlorinated dibenzo-p-dioxins and dibenzofurans (PCDDs/Fs) and dioxin-

like polychlorinated biphenyls (dl-PCBs) sample preparation. Samples from 2011 – 2012 were prepared as follows: the concentrated extracts underwent clean-up using a sulphuric acid ($H_2SO_4$) modified silica column, eluted with 40 mL DCM/n-hexane mixture (1:1). Fractionation was achieved in a disposable Pasteur pipette micro column containing (from bottom to top): 50 mg silica, 70 mg charcoal/silica (1:40) and 50 mg of silica (Darco G60 charcoal). The column was prewashed with 5 mL of toluene, followed by 5 mL of DCM/cyclohexane mixture (30%), then the sample was loaded and eluted with 9 mL

DCM/cyclohexane mixture (30%) in fraction 1 (mono-ortho dl-PCBs, PBDEs) and 40 mL of toluene in fraction 2 (PCDDs/Fs, non-ortho dl-PCBs). Each fraction was concentrated to the final volume of 50 µL and transferred into an insert in a vial. Samples from 2013 – 2014 were prepared as follows: the clean-up column was achieved using a multi-layer silica column (KOH silica, $H_2SO_4$ silica, $Na_2SO_4$, prewashed with *n*-hexane), analytes were eluted with 120 mL of *n*-hexane. Fractionation was performed on a carbon column packed with 50 mg of AX-21 active carbon dispersed on 1 g of Celite 545. After elution

with 18 ml of a mixture of cyclohexane-DCM-methanol (2:2:1, v/v) (fraction 1, part of ortho PBDEs) and 6.5 ml of toluene (fraction2 non-ortho PBDEs, dl-PCBs), fraction 3 eluted with 80 ml of toluene applied on a column with reverse flow was



collected (PCDDs/Fs). After instrumental analyses of dl-PCBs, fraction 1 and 2 were combined, transferred to an insert in a vial, spiked with recovery standard ($^{13}$C BDE-77 and 138) and analysed for PBDEs.

Ten PBDEs (BDE-28, BDE-47, BDE-66, BDE-100, BDE-99, BDE-85, BDE-154, BDE-153, BDE-183 and BDE-209) were analysed using high resolution on an Agilent 7890A GC (Agilent, USA) equipped with a 15 m × 0.25 mm × 0.10 µm DB-5 column (Agilent, J&W, USA) (samples from 2011) or a 15 m × 0.25 mm × 0.10 µm RTX-1614 column (Restek, USA) (samples since 2012 onwards) coupled to an AutoSpec Premier MS (Waters, Micromass, UK). The MS was operated in EI+ at the resolution of >10000. The temperature programme was 80°C (1 min hold), then 20°C min$^{-1}$ to 250°C, followed by 1.5°C min$^{-1}$ to 260°C and 25°C min$^{-1}$ to 320°C (4.5 min hold). The injection volume was 3 µL in splitless mode at 280°C, with He used as a carrier gas at constant flow of 1 mL min$^{-1}$. The instrumental limits of quantification (iLOQs) were determined from calibration curves or from individual sample chromatograms corresponding to a signal-to-noise ratio > 9.

### 2.3 QA-QC

Eleven field blanks and eleven laboratory blanks were analysed as per samples. Except for BDE-209, blank levels of individual analytes were below detection or low otherwise (on average <5% of sample mass for detected compounds), suggesting minor contamination during sampling, transport and analysis. In case of BDE-209, high blank levels were found in some cases (on average 10.1% and 35.1% of sample mass for GFF and PUF, respectively). The higher blanks are probably caused by elevated background concentrations of BDE-209 which may be related to the microabrasion of particles from plastic material containing BDE-209 (Webster et al., 2009). The PBDE concentrations presented here have been blank corrected by subtracting the average of the field blanks, separately for GFFs and PUFs. The PBDEs were quantified using isotope dilution and thus recovery-adjusted. Mean PBDE recoveries (± standard deviation) ranged from 60.9% for BDE183 to 149.9% for BDE209 with an average value of 92.7%. Limits of quantifications (LOQs) were determined as the maximum of the iLOQs and the average of the field blanks plus three times their standard deviations. LOQs ranged from 2.09E-05 to 1.04 pg m$^{-3}$ (Table S2).

### 2.3 Modelling of gas-particle partitioning

Partitioning of organic compounds such as PBDEs between the gas and particle phases is often described using the gas-particle partition coefficient, $K_p$ (in m$^3$ µg$^{-1}$) defined by Harner and Bidleman (1998) as:

$$K_p = (C_p/C_{TSP})/C_g \tag{1}$$

where $C_p$ and $C_g$ are the concentrations of individual PBDEs (in pg m$^{-3}$) in the particulate and gaseous phases, respectively and $C_{TSP}$ is the concentration of the total suspended particles (TSP) in the air (in µg m$^{-3}$).

Accurate knowledge of $K_p$ is crucial for modelling the fate of PBDEs in the environment. In this study, we compared the experimental $K_p$ values with those determined by three predictive models. For comparison, we considered only cases where individual PBDEs were detected in both the gas and the particle phase. The first approach, also known as the $K_{OA}$ model,



predicts $K_p$ based on the octanol-air partition coefficient ($K_{OA}$). It implicitly assumes that equilibrium has been reached between the two phases and that absorption into particulate organic matter (OM) of the particles determines the distribution process, while other types of molecular interaction (i.e. adsorption to minerals or soot) are neglected (Harner and Bidleman, 1998a). Then, assuming that the activity coefficient of the absorbing compound and its molecular weight is the same in octanol and

organic matter, $K_p$ can be defined as (Harner and Bidleman, 1998b):

$$\log K_{pe, abs} = \log K_{OA} + \log f_{OM} - 11.91 \tag{2}$$

where the subscript e,abs in $K_p$ highlights the equilibrium assumption of this approach and the fact that it considers only absorptive contributions and $f_{OM}$ is the fraction of organic matter phase on particles.

The second approach used is the steady state model proposed by Li et al., (2015) in which $K_p$ is defined as:

$$\log K_{ps, ss} = \log K_{pe, abs} + \log \alpha \tag{3}$$

where log α represents the non-equilibrium term due to disturbances from wet and dry deposition of particles and is defined as:

$$\log \alpha = - \log(1 + G/C) \tag{4}$$

where C = 5 and G = 2.09 x $10^{-10}$ $f_{om} K_{OA}$ (5)

Finally, the last approach is based on the quantitative structure-property relationship (QSPR) model recently proposed by Wei et al., (2017). To fit this regression model, several properties were calculated quantum mechanically for each PBDE molecule in the gas phase. The regression fitting was done for a dataset where temperature varied between 10 and 32 ºC. This model also implicitly assumes that equilibrium exists between PBDEs in the gas and particle phase. In this approach, $\log K_p$ is defined as:

$$\log K_{pe, QSPR} = 0.026 \, V -0.030 \, T - 0.858 \, qC^- + 3.864 \, qH^+ +0.002 \tag{6}$$

where V is the molecular volume, T is the ambient temperature, $qC^-$ is the most negative charge on a carbon atom and $qH^+$ is the most positive charge on a hydrogen atom (Wei et al; 2017). These were Mulliken charges calculated with density functional theory at the B3LYP/6-31G(d,p) level.

Equations (1-6), can be used to predict the particulate fractions ($\theta_{pr}$) using:

$$\theta_{pr} = K_p C_{TSP}/(1+K_p C_{TSP}) \tag{7}$$

Given that most PBDEs are sorbed to fine particles (Okonski et al., 2014), the concentration of particles smaller than 10 µm ($PM_{10}$) instead of $C_{TSP}$ and the measured $f_{OM}$ at this site were used (data provided by the Czech Hydrometeorological Institute, http://www.chmi.cz). The temperature dependence of $K_{OA}$ for all PBDEs, except BDE209, was determined from published relationships, based on direct measurements (Harner and Shoeib, 2002). Given the uncertainties while estimating such important physicochemical properties as $K_{OA}$ from other parameters, we did not consider BDE209 in the predictions.

2.5 Meteorological data and air mass origin

Continuous meteorological data, 2-m temperature, relative humidity (RH), 2-m wind speed and direction were provided by the observatory (Czech Hydrometeorological Institute).



The influence of LRAT was assessed by evaluating the backward trajectories of specific samples. Indeed, the Lagrangian particle dispersion model FLEXPART (Stohl et al., 2005) was used to identify air mass origins of the ten samples showing the highest and the lowest PBDE concentrations from our dataset. The meteorological data (0.5° and 3 hours resolution, 91/137 vertical levels) were retrieved from the ECMWF database (http://www.ecmwf.int). For every weekly sample investigated, 100 000 particles were released between 0 and 200 m agl and were followed 5 days backward in time. Additional details can be found elsewhere (Mulder et al., 2015).

## 3 Results

### 3.1 Breakthrough and sampling artefacts

Breakthrough is an issue of concern as relatively high sample volumes (i.e. >1000 m$^3$) are usually used to quantify trace contaminants such as PBDEs. Breakthrough of gas-phase PBDEs was evaluated by quantifying separately each of the two PUFs placed in series for all the weekly air samples collected in 2012 (N = 25, sampled volume = 4015–5864 m$^3$). This covered a large range of meteorological conditions and the results are considered applicable to the other years. Summary of the results of the breakthrough experiments is presented in Table S3 and Figure S1. On average, less than 20% of individual PBDEs was found on the lower PUF, except for BDE183 and BDE209 (Table S3). Based on these results, the current sampling configuration with two PUFs in series is considered to be efficient for trapping these PBDEs in the gas phase although the reported gaseous concentrations for these congeners may be underestimated by up to 4%.

In case of BDE183 and BDE209, on average 31.9 and 53.6 % were found on the lower PUF, respectively, in agreement with a more detailed breakthrough study previously published (Melymuk et al., 2016a). Given that these compounds are not volatile (i.e. vapor pressure of 3.30 10$^{-6}$ and 1.43 10$^{-8}$ Pa, respectively, Yue and Li, 2013), it is unclear what causes the high fractions found on the lower PUFs. We suggest a few possible reasons for these unexpected results. Firstly, this could be due to the uncertainties and limits with the analytical quantification, associated with potential contamination in the laboratory. This is particularly true for BDE209 characterized by higher LOQs and higher blank levels (see Sect. 2.3) and considering that the analysis of this congener is more challenging (Law et al., 2008), but not for BDE183. Secondly, this could be due to blowoff, which is the volatilisation loss of SOCs from the filter, thereby disproportionately increasing the SOC PUF masses (Melymuk et al., 2014). However, it is unknown to which extent this possible sampling artefact may occur although it has been reported in different studies (Allen et al., 2007; Besis and Samara, 2012). Finally, it is unclear whether some flame retardants (likely the Deca mixture) have been used in some electronic and plastic parts present within the air sampler, contaminating the PUFs. Based on the results of the breakthrough evaluation, we can conclude that the sampling set-up was deemed appropriate for the quantification of all PBDEs, except for BDE183 and BDE209 for which some uncertainty is associated with the reported gaseous concentrations and particulate fractions.



## 3.2 PBDE concentration levels

Except BDE66 and BDE85, all congeners were detected in >88% of the samples (Table S4), highlighting their persistency in the environment. In this study, the total (gas and particles) concentrations of $\Sigma_9$PBDEs (all congeners except BDE209) ranged

from 0.0882 to 6.08 pg m$^{-3}$ with an average value of 0.542 pg m$^{-3}$. BDE209 had a lower average concentration of 0.468 pg m$^{-3}$ (ranging from <LOQ to 5.01 pg m$^{-3}$) (Table S5). The PBDE concentrations reported here were similar to those observed for other European background or remote sites (Degrendele et al., 2016; Iacovidou et al., 2009), which are usually lower than 5 pg m$^{-3}$ (Lee et al., 2004). These background levels are lower than those previously reported for urban sites (Moeckel et al., 2010; Okonski et al., 2014; Salamova and Hites, 2011) and particularly in indoor air (Besis and Samara, 2012; Melymuk et

al., 2016b).

Besides BDE209 which contributed on average to 46.3% of all PBDEs measured, BDE47, 99 and 183 showed the highest concentrations accounting on average for 31.5%, 24.3% and 14.1% of $\Sigma_9$PBDEs, respectively. As observed in Figure S2, the PBDE profile differed between the two atmospheric phases with the light congeners having a larger contribution to $\Sigma_9$PBDEs in the gaseous phase compared to the particulate phase. This PBDE profile, with BDE209 being the prevalent congener, is

typical for European environments (Besis et al., 2017; Besis and Samara, 2012; Okonski et al., 2014), while BDE47 and BDE99 usually dominates the BDE levels in North America (Besis and Samara, 2012; Hoh and Hites, 2005; Strandberg et al., 2001). Though, given that the technical Deca mixture covered about 83% of the global PBDEs market (in 2001; Besis and Samara, 2012), these distributions suggest that lower congeners are more prone to volatilisation from products or from other environmental media compared to the higher brominated congeners or that photolytic degradation of BDE209 to lower

brominated BDEs is occurring (Luo et al., 2014). Indeed, an increase of lower brominated congeners (Hexa-through Nona) was observed under photolysis of BDE209 in solvents, sediments, soils and sands (Eriksson et al., 2004; Söderström et al., 2004). However, photolysis is not specific to BDE209 but relevant for all congeners, such as e.g., BDE99 (formation of BDE47; Fang et al., 2008; Sanchez-Prado et al., 2005).

## 25   3.3 Factors affecting the inter sample variations

Different parameters can influence the seasonality of PBDE atmospheric concentrations such as the advection from urban and industrial sources, the efficiency of removal processes (degradation and deposition) and the meteorological conditions (e.g. temperature, boundary layer height, precipitation). The results of the correlation analysis between the gaseous, particulate and total concentrations of individual PBDEs and different meteorological parameters are shown in Table S5.

No or low influence of wind speed and wind direction on the PBDE concentrations were observed, consistent with the type of site (background) and in agreement with previous studies (Besis et al., 2015; Cetin and Odabasi, 2008). As observed elsewhere (Dien et al., 2015), the particulate concentration of high brominated PBDE (i.e. 100, 99, 154, 153, 183 and 209) were negatively





correlated to the precipitation rate. This confirms the significant washout of congeners partitioning mostly to the particulate phase compared to those in the gas-phase (Venier and Hites, 2008a), a general trend for lipophilic organic compounds (Ligocki et al., 1985; Shahpoury et al., 2015). Moreover, the RH is suggested to have a significant effect on the particulate phase concentrations of all PBDEs, except BDE28 and BDE209, as well as a negative influence on some gaseous PBDEs

concentrations (Table S5). No conclusive trends can be expected as a consequence of the association of RH with precipitation and temperature, and these parameters' influences on scavenging and gas-particle partitioning, largely varying across types of scavenging (in-cloud, below cloud), precipitation, and aerosols. The atmospheric boundary layer (ABL) height shows strong correlations with the particulate concentrations of all PBDEs except BDE28, in agreement with a previous study (Dien et al., 2017). The ABL height was also shown to be a primary driver of PBDE concentration's diel variability (Moeckel et al., 2010).

In this study, when considering the total concentrations of individual PBDEs, a significant influence of ambient temperature was suggested only for BDE47 and BDE66 (higher concentrations for higher temperatures) and BDE153, BDE154 and BDE183 (higher concentrations for lower temperatures) (Table S5). On the other hand, no significant seasonality was observed for the remaining congeners. This is in contradiction with many previous studies which reported higher concentrations of most PBDEs with the exception of BDE209 in summer compared to winter (e.g. Birgul et al., 2012; Cetin and Odabasi, 2008).

Another study over Japan (Dien et al., 2015) found higher concentrations of lower brominated congeners (BDE47 and BDE99) in warm season while those of higher brominated congeners mainly bound to particles (e.g. BDE183 and BDE209) peaked in winter, similarly to this study. Overall, the absence of seasonality in the total concentrations of most PBDEs suggests that their atmospheric levels are still driven by primary sources. Different behaviour found in the Mediterranean (Birgul et al., 2012; Cetin and Odabasi, 2008) might be related to higher temperatures there, throughout all seasons. However, the effect of ambient

temperature on the particulate and gaseous concentrations of individual PBDEs was evident and shed lights on the processes that drive partitioning behaviour. Indeed, the particulate concentrations of all individual PBDEs were significantly ($p<0.05$) higher at colder temperatures (Figure S3), as found for semivolatile organics in general (Bidleman, 1988). This is furthermore in agreement with a previous study conducted at a rural and an urban site in the Czech Republic where higher particulate PBDEs concentrations were also found in winter, and this was attributted to temperature-induced shifts in gas-particle

partitioning (Okonski et al., 2014). The higher degradation in summer and lower ABL height in winter may also support higher particulate PBDEs at cold temperatures. Additionally, Lee et al., (2004) proposed that low ambient air temperatures may cause increased emissions of PBDEs from anthropogenic activities such as combustion.

Correspondingly, the gaseous concentrations of all PBDEs, with the exception of BDE28, BDE66 and BDE209, significantly ($p<0.05$) increased with ambient temperature (Table S5, Figure S3), consistent with previous studies (Melymuk et al., 2012;

Venier and Hites, 2008b). This is indicative of the influence of air-surface exchange (i.e. revolatilisation from soils) on the ambient PBDE gas-phase concentrations. Similarly to this study, Ma et al. (2013) also found that the gaseous concentrations of BDE47 at different sites around the Great Lakes was maximal in summer, while the particulate ones peaked in winter. In case of BDE209, some previous studies reported, as here, no seasonality (Cetin and Odabasi, 2008; Su et al., 2009).



It is interesting to note the absence of influence of ambient temperature on the gaseous concentrations of BDE28 and BDE66 (Table S5, Figure S3), as these compounds are mainly present in the gas-phase and are volatile enough to undergo air-surface exchange processes. We suggest that there is another process, not necessarily related to ambient temperature, that controls the gaseous concentrations of these congeners. This could be the photolytic debromination of higher brominated congeners

(Bezares-Cruz et al., 2004; Wei et al., 2013).

Air masses mainly originating from the West, South-West or North-West, i.e. air that has passed through the Atlantic Ocean or the North Sea, were found for 8 of the 10 samples with the lowest PBDE concentrations (Figure S4). In contrast, the samples with $\Sigma_9$PBDEs > 1 pg m$^{-3}$ were not associated with air masses from a clear direction but rather by air that stagnated over continental Europe (Figure S5). The fact that the highest PBDE concentrations were observed under advection from different

directions suggests that there is a rather homogeneous continental emission source. The high PBDE concentrations observed in these samples is likely due to short and intense emissions of flame retardants, as for example during the incineration of products or waste containing PBDEs.

To conclude this section, the atmospheric concentrations of individual PBDEs were controlled by primary emissions (including combustion and evaporation), re-volatilisation from soils, deposition processes (rain scavenging) and LRAT.

**3.2 Gas-particle partitioning in air samples**

Results of PBDE gas-particle partitioning from individual studies at a global scale are contradictory. For example, some studies have found that most PBDEs have small particulate fractions (Besis et al., 2017; Iacovidou et al., 2009; Mandalakis et al., 2009; Mulder et al., 2015) while other studies found that, for a specific temperature, the particle fraction significantly increased with increasing degree of bromination (Chen et al., 2006; Davie-Martin et al., 2016; Möller et al., 2011; Strandberg et al.,

2001). In this study, with the exception of BDE28 and BDE209, which were detected in about half of the samples only in one phase (Table S4), the remaining congeners were significantly detected in both phases. The particulate fraction ($\theta_{measured}$) significantly increased with the degree of bromination (Figure S6). For example, for BDE28, the average $\theta_{measured}$ was 0.11, while it was 0.23 for BDE47, 0.49 for BDE99, 0.62 for BDE154 and 0.71 for BDE183 (Figure S6). This is consistent with previous studies (Davie-Martin et al., 2016; Strandberg et al., 2001). However, it is important to note that large seasonal

variations were observed (Figure 1 and S6). Indeed, while $\theta_{measured}$ of BDE47 was on average 0.51 in winter, this was only 0.01 in summer. Similarly, for BDE99, $\theta_{measured}$ was 0.88 and 0.11 in winter and summer, respectively. Statistically significant ($p<0.05$) correlations between $\theta_{measured}$ and $1/T$ was found for all individual congeners investigated (Table S6). These important seasonal variations in the gas-particle partitioning of PBDEs have been previously reported but to a lower extent than in the present study (Davie-Martin et al., 2016; Su et al., 2009). This finding implies that the temperature is an important variable,

obviously affecting the partitioning of PBDEs in the atmosphere. These seasonal variations in gas-particle partitioning must be taken into account when both considering the LRAT potential of PBDEs or developing environmental models (independent of spatial scale).



In the case of BDE209, the reported particulate fractions may be associated with uncertainties (see Sect. 3.1). This congener was found in about half of the samples only in the particulate phase and the average $\theta_{measured}$ was 0.74. The influence of ambient temperature on the particulate fraction of BDE209 was statistically significant ($p<0.05$) but less pronounced than for the other congeners (Table S6). Previously reported particulate fractions for this compound ranged between extreme values (i.e. $\theta = 0$-

1) (Cetin and Odabasi, 2007), though some studies reported it mainly in the particulate phase (Cetin and Odabasi, 2008; Li et al., 2016; Ma et al., 2013; Strandberg et al., 2001; Su et al., 2009), and others mainly in the gas phase (Agrell et al., 2004; and references within Li et al., 2016). Li et al., (2016) recently reported on BDE209 levels found on a global scale and also noted the large range of particulate fractions reported.

**3.5 Modelling of gas-particle partitioning**

As presented in Figures 2 and S7, none of the three model approaches successfully predicted $K_p$ or $\theta$ for all individual PBDEs considered. The $K_{OA}$-model generally captured the overall trend regarding seasonal variations of gas-particle partitioning (similar slope as the 1:1 line in Fig 2) but, with the exception of BDE28, consistently overestimated $K_p$ by 1-2 orders of magnitude. This results in an important overestimation of the particulate fraction as, except for BDE28, this model predicted

that the majority of PBDEs would be mostly present in the particulate phase ($\theta_{predicted}$ often $> 0.9$, Fig S7). This is in clear disagreement with our observations. Only for BDE28, this model provided satisfactory results. This overestimation of $K_p$ by the $K_{OA}$-model has been also previously reported for different sites in the Mediterranean and China (Besis et al., 2017; Cetin and Odabasi, 2008; Chen et al., 2006).

Similarly to the $K_{OA}$-model, the estimations provided by the steady state approach were also acceptable only for BDE28. For

the other congeners, this model consistently over- and under-predicted $K_p$ by 1-2 orders of magnitude depending on the compound and season investigated. This model tends to predict that these PBDEs will be within the maximum partition domain (Li et al., 2015) for which $\log K_p$ is constant with a value of -1.53, regardless of the ambient temperature (Fig 2). This model predicted that the maximum particulate mass fraction for all PBDEs would be $\approx 0.6$, given the conditions at the sampling site, which is in disagreement with the observed seasonal variations of this study (discussed above). The only study testing this

model to atmospheric PBDE data did not find an acceptable performance for all PBDEs investigated, though it performed generally better than the $K_{OA}$-model (Besis et al., 2017).

The QSPR model generally tends to underestimate $K_p$ for all compounds studied, except for BDE153 and 183 for which satisfactory predictions were found (Fig 2). For example, for BDE28, this model predicts that at most 7% will be present on particles while in reality, cases with >20% on particles were often found for cold temperatures (Fig S7). We note that this

regression model has been fitted to data within a limited temperature range (10-32 ºC), therefore attempts to extrapolate outside of this range (in this study, the average weekly temperatures were -6.4 to 23.0 ºC) may not be appropriate. However, even within this range, a severe underestimation is seen. We suggest that the complex molecular interactions involved in the



partitioning processes cannot be fully captured based on a limited selection of gas phase atomic charges only. For a truly universal regression model, calculations of the interactions between PBDEs and different particle matrices would be required. As we have seen, none of the models are able to predict the partitioning of PBDEs in a satisfactory way. Though, while considering the average conditions for this study, the overall tendency of predicting $K_p$ or $\theta$ using the steady state or the QSPR

models were higher than those from the $K_{OA}$–model (Figure S8), we do not recommend the use of these models given that the very pronounced seasonal variations observed were not captured. Moreover, we would like to remind that though in most cases, these two models predicted $K_p$ within one order of magnitude of the observed value, this can still result in highly inaccurate values of $\theta$ (Fig S9), therefore are not ideal when phase-specific removal processes are to be estimated.

Addressing unrealistic implicit assumptions of these models is obviously crucial for the understanding of these discrepancies.

The $K_{OA}$-model represents absorption in octanol and therefore does not exactly reflect the true process of adsorption of aerosols (a process that must precede any absorption). We highlight a study by Ding et al., (2014) which investigated the adsorption of different congeners on graphene (a structure that on a molecular level has similarities to black carbon). It was found that, in addition to the number of bromine atoms, the adsorption energy was also affected by the 3-dimensional structure of the PBDE congener. This effect is best illustrated by congeners BDE153 and BDE154, both of which have the same number of bromine

atoms. Steric interactions between bromine atoms in the 6 and 6' positions, such as for BDE154, meant this congener adopted a twisted structure and adsorbed more weakly onto the graphene surface. BDE153 on the other hand can adopt a planar structure and adsorb more strongly; the consequences of this effect are observed in our results (Figure S10). Such effects are not captured by using $K_{OA}$ alone as a predictor (octanol having more degrees of freedom can better accommodate to twisted structures). We speculate these effects could influence the ability of specific BDEs to both adsorb onto and diffuse within the aerosol particle.

Furthermore, octanol is possibly not the perfect surrogate to describe absorption in particulate OM: Better results for prediction of $\theta$ of PAHs were achieved when absorption in octanol was replaced by absorption in two particulate OM phases, using dimethyl sulfoxide and polyurethane, respectively, as the surrogates in a polyparameter linear free energy relationships model (Shahpoury et al., 2016).

Regarding the steady state approach, our results tend to support the concept behind the model, i.e. that equilibrium between

the gaseous and particulate phases is not reached beyond a certain $\log K_{OA}$ (11.5 suggested by Li et al., (2015)). Indeed, we observed a distinctly different behaviour in the gas-particle partitioning for PBDEs with $\log K_{OA} < 11$ within the environmental conditions observed (i.e. BDE28) and for all other PBDEs with $\log K_{OA} > 11$ (Figure S11). Taking into account that the steady state-model considers BDE28 to be within equilibrium (Li et al., 2015) and that the equilibrium $K_{OA}$-model provided satisfactory results only for BDE28, our results tend to suggest that other PBDEs are not within $K_{OA}$-predicted equilibrium but

instead a different equilibrium or steady state. However, it is evident from Figure 2 that the considerations taken within the steady state model are inadequate to correctly characterize the gas-particle partitioning of PBDEs. Li and co-workers (2015) suggested that this deviation from equilibrium is due to the influence of wet and dry deposition. Though we recognize that wet and dry deposition may increase the relative presence of PBDEs in the gas phase, we do not consider this to be the major mechanism resulting in the steady state of most PBDEs for two reasons.



Firstly, this concept should not be exclusive to PBDEs but should also be valid for other SOCs such as benzo(a)pyrene, a high molecular weight polycyclic aromatic hydrocarbon (PAH) with a $\log K_{OA}$ of 11.6 at 25°C. This compound is generally found only in the particulate phase (Shahpoury et al., 2015) with only limited amount in the gaseous phase. It is therefore unclear why disturbances due to wet and dry deposition should be more pronounced for PBDEs than PAHs. Secondly, we recognize

that because rain scavenging is more efficient for particles than gases (Wania et al., 1998), samples associated with more intense precipitation are likely to have a lower particulate fraction. This overall trend was observed in this study as statistically significant ($p<0.05$) correlations between $\theta_{measured}$ and the precipitation rate were present for all PBDEs, with only the exception of BDE85 and BDE209 (Table S7). However, the sample with the highest precipitation rate (i.e. 112 mm) had a higher particulate fractions than the following sample which had almost no rain (i.e. 0.4 mm) and this was observed also for other

subsequent samples. Therefore, we do not consider wet deposition (nor dry deposition) to be the factor governing the equilibrium (or absence it) of PBDEs. As previously suggested by Cetin and Odabasi, we consider that the higher presence of PBDEs in the gas phase (compared to that expected based on $K_{OA}$), is due to their departure from equilibrium partitioning and that the relaxation to equilibrium is slower for compounds with higher $\log K_{OA}$ (Cetin and Odabasi, 2008).

To look more widely at processes that could cause departure from the $K_{OA}$ predicted equilibrium, we should also recognize

that there are other factors, beyond the thermodynamic stability of PBDEs in the particle phase, which could also influence the particulate fraction. We cannot assume the lifetime of PBDEs in the particle phase is identical to the lifetime in the gas phase. If the difference between these two lifetimes becomes significant, we would expect a shift from the $K_{OA}$ predicted equilibrium. Li et al., (2015) considered this idea in terms of dry and wet deposition. We suggest there may also be chemical factors that influence this process. We note a study by Raff and Hites, (2007) where gas phase photolysis rate constants are estimated for

different BDE congeners. Even amongst congeners with the same number of bromine atoms, significant differences exist in gas phase lifetimes, for example between BDE99 (4 hours) and BDE100 (54 hours).

### 3.2 Inter-annual variations

Several years following the inclusion of the Penta- and Octa-BDE mixtures to the Stockholm Convention, long-term data can be used to assess whether the environmental levels are decreasing as a consequence of primary emissions reduction. The

atmosphere is an ideal environmental compartment as it is particularly responsive to changes in primary emissions (Harrad, 2015).

To evaluate whether the atmospheric concentrations of individual PBDEs were significantly declining or not, the atmospheric concentrations of individual PBDEs ($C_i$, in pg m$^{-3}$) were plotted assuming first order kinetics using the formula:

$$\ln C_i = \ln C_{0,i} - k_i t \tag{8}$$

Where $C_{0,i}$ is the theoretical concentrations of individual PBDE measured at $t_0$ (i.e. the end date of PBDE production), k is the rate constant and t the time in days. The apparent halving time ($\tau_{1/2}$) describes the time period it takes to reduce/increase the initial PBDE concentrations to half/twice its value and should not be confused with half-lives related to degradation processes. It was calculated from k as:



$\tau_{1/2} = \ln 2/k$ (9)

By applying Eq (9) to all samples, significant decreases at the 95% confidence interval were found for BDE100, BDE99, BDE153 and BDE209 with apparent half-lives of 2.83, 3.61, 4.52 and 2.58 years, respectively (Figure 3 and Table S8) but not for the remaining congeners.

Previous research performed on long-term trends of PBDEs in the atmosphere are available mainly for UK and North America. Indeed, at different UK and Norwegian background sites, Schuster et al; (2010) reported significant decreases at four of the eleven sites investigated in 2000-2008 of BDE47, BDE100, BDE99, BDE153 and BDE154 with half-lives of 1.4-4.0 years. At two urban sites in UK, significant decreases were also found for $\Sigma_9$PBDEs with half-lives of 2.0-3.4 years (Birgul et al., 2012). Other studies also reported significant decrease of PBDEs at three UK sites in 2000-2010 (Graf et al., 2016) but also in

Japan in 2009-2012 (Dien et al., 2015). On the other hand, at five sites around the Great Lakes and at a rural site in the UK, no clear and consistent decline in PBDEs concentrations were found (Birgul et al., 2012; Ma et al., 2013).

Overall, the results from the present study tend to show that the primary emissions of BDE99, 100, 153 and 209 are declining in Central Europe. The similar rate of declines observed for other European sites (Table S8) indicates that regional scale primary emissions are controlling the trends (Schuster et al., 2010). This is the first study reporting significant decreases only

for some of the high brominated congeners but not for the low ones., which have lower first order removal rates (Wei et al., 2013). However, though decreasing trends of some congeners are observed in different locations worldwide, we should keep in mind that PBDEs are still persisting in the environment and that a time lag is needed to clearly see the effect of reduction in primary emissions on background atmospheric concentrations of all PBDEs (Ma et al., 2013).

Interestingly, BDE28 and BDE66 showed an overall increasing trend, although this was not statistically significant (Figure

S12 and Table S8). We can note that statistically significant increasing trends were observed when considering only the autumn and summer samples for BDE28 and BDE66, respectively, with doubling times of 4.23 and 1.44 years, respectively. Similarly, in the Great Lakes area (US), Ma et al., (2013) found that the gaseous concentrations of BDE47 and BDE99 were significantly increasing from 2005 to 2011 at three rural/remote sites with longer doubling times for BDE47 (7-9.4 years) compared to BDE99 (4.3-4.7 years) (Table S8). Taking this into account and considering that BDE28 and BDE66 are products of the

debromination of higher BDE congeners (Vesely et al., 2015; Wei et al., 2013) this is an additional indication that photolytic degradation of higher to lower brominated congeners is occuring in the atmosphere. Results from a modelling study concluded that 13% of the Penta-BDE occuring in the environment resulted from the degradation of Deca-BDE induced by photolysis (Schenker et al., 2008). The authors argued that once the Penta mixture would be phased out completely, the importance of Deca-BDE as a source of Penta-BDE will increase. Here, we would argue that over the next decades, an increase or a steady

state in the atmospheric concentrations of low brominated PBDEs may occur and that the congener profile will likely be dominated by those lighter congeners which are more prone to re-volatilisation (and have a higher persistency) and hence have a higher potential for long-range atmospheric transport. The fact that, in this study, no significant decrease was observed for BDE47 in comparison to BDE99 while it originates from the same Penta mixture and is known to be a debromination product



of BDE99 (Bezares-Cruz et al., 2004) support this hypothesis. However, monitoring air concentrations over longer time span is needed to provide further evidence.

## 4 Conclusions

This study has shown that the atmospheric PBDE levels are governed by primary emissions, re-volatilisation from soils,
deposition processes and LRAT.

One important finding of this study is the seasonal variations of the particulate fraction which were observed for most PBDEs. This has implications for studies using passive sampling design for which the efficiency of particulate collection is still uncertain. Therefore, the interpretation of the seasonal variations of PBDEs from such studies should be done in a cautious manner. Moreover, one should keep in mind that the congener profiles observed in this study differed between the gaseous and
the particulate phase. Therefore, using a sampler collecting only one specific phase would provide a different congener profile. Moreover, this study has shown that, at the current state of knowledge, none of the available models was able to effectively characterise the gas-particle partitioning of PBDEs. Though some of the tested models provided acceptable predictions for some of the compounds, none was satisfactory for all PBDEs investigated and for the conditions met at this sampling site. This highlights the need for a gas-particle partitioning scheme for PBDEs that would be universally applicable under a range of
atmospheric conditions. This is the minimal criteria to be able to adequately characterize the environmental fate of PBDEs at a global scale.

Finally, the results from this study tend to show that the debromination from high to low brominated congeners, enhanced by photolysis, is also an important process governing PBDE concentrations in the atmosphere. Given that nowadays, all formulations have been phased out, we may expect an enrichment in light congeners in the environment at a global scale. As
these compounds are more volatile and have higher persistency than heavier congeners, their secondary formation enhanced by photolysis may be a serious issue of concern. Further studies should confirm whether the atmospheric concentrations of lower brominated PBDEs will increase or be at a steady state within the next decades.

## Supporting information

Description of samples collected, results of the breakthrough analysis, summary of individual PBDE concentrations and results of the correlation analyses are provided.

## Acknowledgements

This work was carried out with the support of National sustainability programme of the Ministry of Education, Youth and
Sports (MEYS) of the Czech Republic (LO1214) and the RECETOX (LM2015051) and ACTRIS (LM2015037) research infrastructures funded by the MEYS and the European Structural and Investment Funds (CZ.02.1.01/0.0/0.0/16_013/0001761



and CZ.02.1.01/0.0/0.0/16_013/0001315). The authors are thankful to Roman Prokeš (MU) for the field measurements, Jiří Kalina (MU) for help with statistics and Milan Váňa (Czech Hydrometeorological Institute) for supporting data.

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

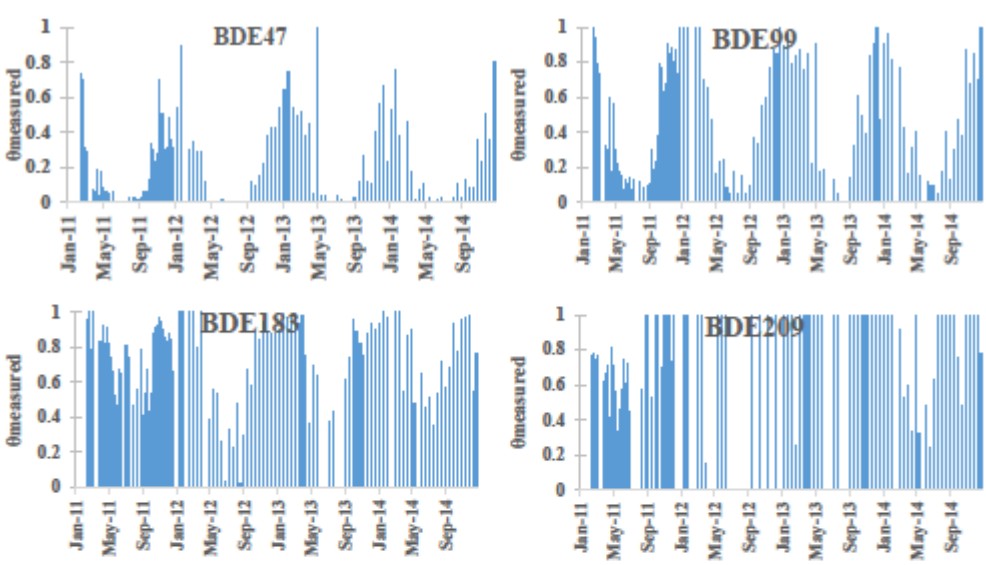

**Figure 1: Measured particulate fraction ($\theta_{measured}$) of selected PBDEs.**





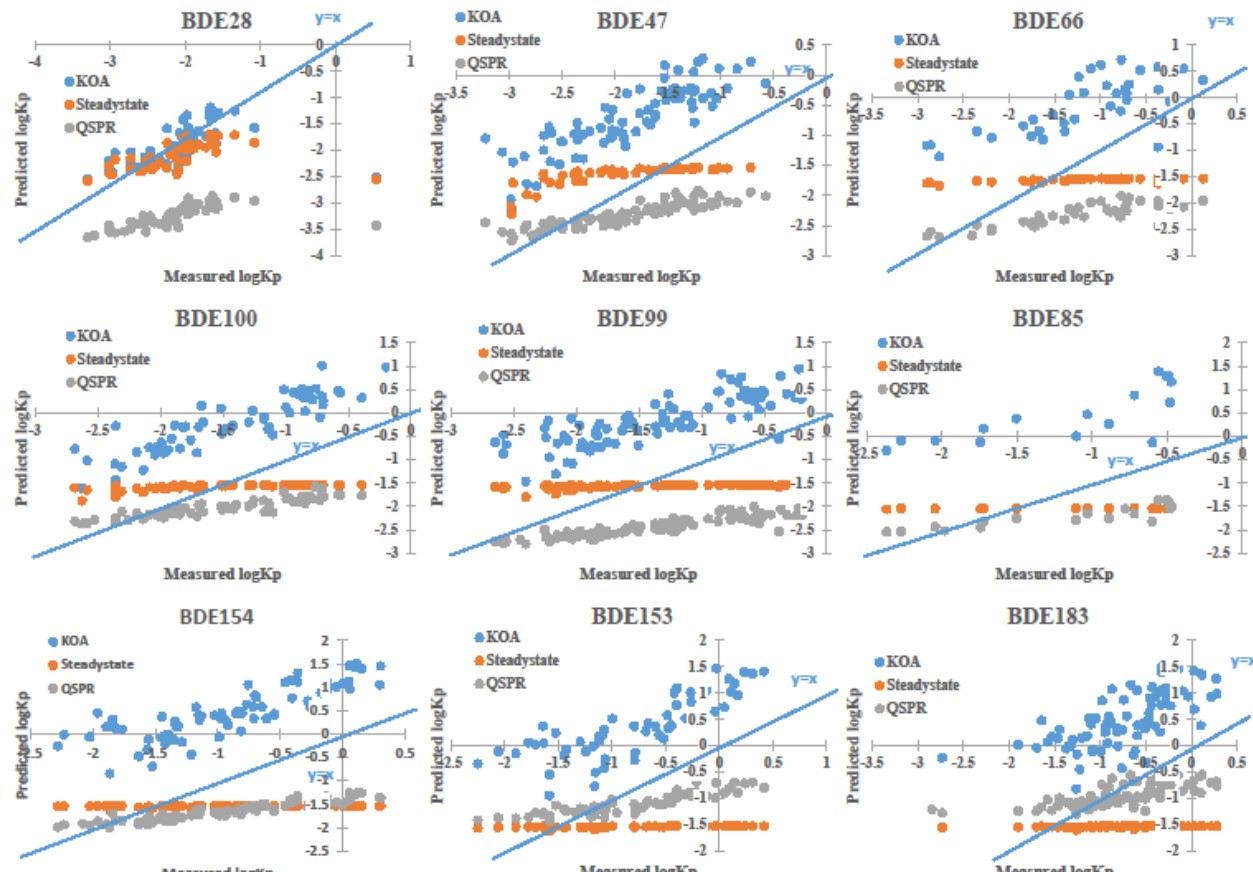

**Figure 2: Comparison of measured and predicted log$K_p$ of individual PBDEs.**







**Figure 3: Multi-year trends of the most abundant PBDEs.**