# Peer review of "Are atmospheric PBDE levels declining in Central Europe? Examination of the seasonal and semi-long term variations, gasparticle partitioning and implications for long-range atmospheric transport"

_Atmospheric Chemistry and Physics, 2018_

## Referee Comment (RC1) · Anonymous Referee #1 · 18 Apr 2018

MS No.: acp-2018-144 Title: Are atmospheric PBDE levels declining in Central Europe? Examination of the seasonal variations, gas-particle partitioning and implications for long-range atmospheric transport Author(s): Céline Degrendele et al. MS Type: Research article

General comments The manuscript presents four-year monitoring data (2011-2014) on atmospheric polybrominated diphenyl ethers (PBDEs) at the Košetice observatory, in an agricultural region in central Czech Republic. Particle- and gas-phase samples were collected on a weekly basis (7-day sampling duration) using a high-volume air

sampler with PM10 pre-separator equipped with QFFs and 2 PUF plugs in series. PBDEs analysis was performed on 101 valid weekly samples (31 from 2011, 25 from 2012, 20 from 2013, 25 from 2014). Correlations were examined between the individual concentrations (g, p, g+p) of PBDEs and prevailing meteorological parameters. The g/p partitioning of PBDEs, with the exception of BDE209, was investigated by comparing experimental partition coefficient, Kp, values with those determined by three predictive models, the KOA model, a steady state model proposed in literature, and a regression model based on the quantitative structure-property relationship (QSPR) also proposed by other investigators. The apparent halving times ($\tau 1/2$) were calculated to investigate potential declining trends. The LRAT was also assessed by evaluating the backward trajectories of a small number of samples (10) using the Lagrangian particle dispersion model FLEXPART.

The authors have used appropriate methods for sampling/analysis of PBDEs and a thorough QA-QC procedure. The manuscript contains interesting data concerning the g/p partitioning behavior of PBDEs and the semi-longterm trends of their atmospheric levels at a background area of Central Europe.

My major concern is the large and variable amounts found for many PBDEs in the 2nd PUF plug. Since these amounts were included in the gas phase fraction probably resulted to underestimation of the particle fraction $\theta$measured.

Another question is why the subcooled-liquid–vapor pressure (PL)-based model was excluded from the g/p partitioning analysis.

The exclusion of BDE209 from all g/p partitioning models needs explanation. In addition to the above deficiencies, the manuscript needs substantial revision concerning various obscurities, inconsistencies, lacking information data, and missing references in the reference list. Finally, there is much room for language use improvement.

Specific comments Title: The manuscript does not provide information on the seasonal variations of PBDEs levels, therefore "seasonal variations" in the title shall be replaced

by "semi-longterm variations".

2.2 Sample preparation and analysis It is obscure here whether the authors used a different preparation procedure for samples collected in 2013 and 2014 than those used for samples from 2011 – 2012. Please, clarify. 2.3 Modelling of gas-particle partitioning P.5. L. 27: the measured fOM value for this site shall be provided. P.5. L. 29-30: The statement "Given the uncertainties while estimating such important physicochemical properties as KOA from other parameters, we did not consider BDE209 in the predictions" needs further clarification. Do the authors mean that the uncertainties for estimating KOA from other parameters is larger for BDE209 than for the lower PBDE congeners? In any case, the exclusion of BDE209 from all g/p partitioning models shall be explained.

3.1 Breakthrough and sampling artefacts In the breakthrough experiments on the 25 samples from 2012, a significant amount of PBDEs was found in the lower PUF plug, particularly for BDE183 and BDE209 (on average 31.9% and 53.6 % of their total gas-phase concentrations, respectively). • Possible contamination of the PUFs from the electronic/plastic parts of the air sampler is considered as one of the reasons. However, in such a case, the contamination level would be the same in each sampling. Did the authors check that? • The authors say that these findings are in agreement with a detailed breakthrough study previously published (Melymuk et al., 2016a), however this reference is not in the reference list. Also, in their explanation for possible volatilisation loss from the filter, they cite Melymuk et al., 2014, which is also missing from the reference list. 3.2 PBDE concentration levels P.7, L. 5: The average gas- and particle-phase concentrations of BDE209 provided in Table S5 (0.513 and 0.257 pg m-3, respectively) seem to be in discrepancy with the average measured particulate fraction ($\theta$measured) presented in Figure S6, which ranges between 55-85% in the four seasons. Please, check and correct if needed. P.7, L. 7: Degrendele et al., 2016 cited here is missing from the reference list. Please, provide it. Besis et al., 2017 could also be cited at this point as providing PBDEs concentrations at background sites in Europe.

• P.7, L. 10: Melymuk et al., 2016b is not in the reference list. • P.7, L. 9, 15, 16, 18: Besis and Samara, 2012 is not in the reference list. Actually, Besis and Samara 2012 is not dealing with the g/p partitioning of PBDEs. Perhaps the authors wanted to cite Besis et al., 2016 (Atmospheric occurrence and gas-particle partitioning of PBDEs at industrial, urban and suburban sites of Thessaloniki, northern Greece: Implications for human health, Envir. Poll. 215 (2016) 113-124).

3.3 Factors affecting the inter sample variations • P. 8, L.10-12: The statement "In this study, when considering the total concentrations of individual PBDEs, a significant influence of ambient temperature 10 was suggested only for BDE47 and BDE66 (higher concentrations for higher temperatures) and BDE153, BDE154 and BDE183 (higher concentrations for lower temperatures) (Table S5)"is not true! Table S5 shows negative correlation with 1/T (i.e. positive with T) only for BDE47, while positive for BDEs 66, 153, 154, 183. Please, correct properly. • P.8. L.21: $p > 0.05$ shall be $p < 0.05$ here. • Seasonality is confused here with the correlation with ambient T. Unfortunately, seasonal variations of PBDEs levels are not examined in the manuscript. Correlations with ambient T are as expected. Why the authors did not provide Clausius-Clapeyron plots for the gas-phase concentrations? • The statement in P.8. L. 17-18 "Overall, the absence of seasonality in the total concentrations of most PBDEs suggests that their atmospheric levels are still driven by primary sources." shall be "Overall, the absence of correlation of the total concentrations of most PBDEs with ambient temperature suggests that their atmospheric levels are still driven by primary sources."

3.2 Gas-particle partitioning in air samples

• P.9. L. 28: Again seasonality is confused with the correlation with ambient T. Please, correct properly. • P.9. L. 30: The finding that the temperature is an important variable affecting the partitioning of PBDEs in the atmosphere is not new, it has been shown in all similar studies. The authors could provide the logKp-T relationship as well in addition to the correlation coefficient between $\theta$measured and 1/T. • P.12: I think that the first reason for non considering that the deviation from equilibrium was

due to the influence of wet and dry deposition should be the comparison of the particle fraction of PBDEs between samples with high and low precipitation height. I suggest changing the order of reasons.

3.5 Modelling of gas-particle partitioning âǍć P.10. L. 12: Please change "seasonal" to "temporal". âǍć It would be interesting if the authors showed and discussed the logKp-logPLΣ relationship.

3.2 Inter-annual variations

âǍć The statement "C0,i is the theoretical concentrations of individual PBDE measured at t0 (i.e. the end date of PBDE production)" is not clear. The authors have to further explain if and how they estimated the lnC0,i data used in Eq. (8) and if these data are representative for Central Europe. âǍć Is it sure that T units in Eq. (8) are days and not years? Please, confirm. âǍć It should be clarified that the total (g+p) concentrations were used for Ci in Eq. (8). Why the apparent half-lives were not calculated separately for the two phases?

Conclusions

Supplementary Material âǍć Table S3: For clarity reasons, please change "% of compound mass found on the lower PUF" to "% of gas-phase compound mass found on the lower PUF". âǍć Table S6: Please change title to: "Results of regression analysis between $\theta$measured and the inverse of temperature (K-1) for individual congeners. Numbers in bold indicate cases for which regression coefficients (r2) were statistically significant (p<0.05)" âǍć Table S7: Please change title to: "Results of Pearson correlation analysis between $\theta$measured and the precipitation rate for individual PBDEs. Numbers in bold indicate cases for which the correlations were statistically significant (p<0.05)". âǍć Table S8: Please change title to "Apparent half lives ($\tau$) of individual PBDEs observed in this study and elsewhere…..Âż âǍć Figure S3: Please change legend to: "Correlation between the gaseous concentration of individual PBDEs (ln transformed) with the inverse of temperature".

---

## Referee Comment (RC2) · Anonymous Referee #3 · 20 Apr 2018

This paper presents 4 years of air monitoring data of PBDEs (2011-2014) measured at the background site of Košetice observatory in Central Europe. Gas and particle phases of the air samples taken with a high volume air sampler equipped with a PM10 size-exclusion inlet were analyzed separately. The relationship between meteorological conditions and PBDE air concentrations in the particle phase, in the gas phase and total g+p concentrations were examined. Gas-particle partitioning coefficients (Kp) estimated by three models, namely KOA-based model, steady-state model and a quantitative structure-property relationship (QSPR) model proposed by Wei et al. (2017)

were compared with observed Kp. It was found that none of the models provide satisfactory prediction of the gas-particle partitioning observed for PBDEs measured in ambient air and the authors tried to explain why this is the case. Back trajectories were used to examine potential sources of PBDEs in samples with the highest and the lowest PBDE concentrations. Temporal trends of PBDEs measured were assessed using a simple regression method to estimate the first-order halflives which suggest declining trends. The authors suggest that after PBDEs have been regulated under the Stockholm Convention globally, debromination from higher BDE congeners may result in the enrichment of lower brominated congeners which are more persistent and are more mobile than heavier congeners.

This manuscript presents an interesting and valuable air monitoring dataset of PBDE measured in Central Europe. The authors did a fairly thorough analysis of the gas-particle partitioning observed and relationships of air concentrations with meteorological conditions. I found the fact that none of the three theoretical models provide satisfactory g-p partitioning observed interesting and the authors' attempt to explain why this is the case helpful. However, there are a few issues which I'd like to raise to help improve the manuscript and they are given below.

QA/QC:

Blank correction: Were the sample blank corrected using the annual average of the field blanks or the average of all 4 years of blanks? It is recommendable that the samples be blank corrected with the annual average field blank for 2 reasons: 1. The background levels for PBDEs in the lab may vary over time depending on what was being used and exists in the lab (e.g. old cardboard boxes containing PBDEs etc.); and 2. as this is a long-term air monitoring site, it would be problematic in the future if the samples were not blank corrected using annual average field blank, i.e. after collection of a few more years of samples, the LOQs would change. Consistent data management over the long term is important in generating a consistent dataset for the determination of temporal trends of POPs.

Breakthrough and sampling artefacts:

I am in fact not very concern about breakthrough in the PUF resulting in underestimation of the gas phase concentration. In Bidleman and Tysklind (2018, Chemosphere 192: 267-271), it was demonstrated that when PUF2/PUF1$\leq$0.5, the collected fraction exceeds 90 %. Given the case that <20% of the lower congeners were found on PUF2, most of the PBDEs were probably adequately captured by the PUF1+PUF2 sampling train. I'm more concern about the fact that the maximum percentage of BDE 183 and 209 found on PUF2 was 100%, meaning that nothing was found on PUF1 for some samples. It is impossible that there was 100% breakthrough. This looks more like contamination than breakthrough, especially with BDE 209 which usually has high background levels. P. 6 Line 16, how was this underestimation of "up to 4%" determined? For the reasons given for the "breakthrough" of BDE 183 and 209, the fact that there may be blowoff from the filter should affect all congeners rather than just BDE 183 and 209 alone. It is more a sampling artefact than an explanation for the "breakthrough" observed. If it was blowoff from the filter, one should see BDE 183 and 209 more in PUF1 than in PUF2, i.e. it cannot explain the up to 100 % mass found in PUF2. Also, Okonski et al. (2014) found most of the PBDEs on aerosols <0.95 um and the QFF has a pore size of 2.2 um, have the authors considered fine particles physically breaking through the QFF into the PUF below? With the long sampling duration of 7 days and high flow rate of 31.3 m3/h, physical breakthrough of fine particles is possible. Of course, again this does not explain the high percentage mass found on PUF2. Looking at Table S4, when BDE 209 was detectable in gas phase (although it's not very often 41 %), it seems that its gas phase concentrations were higher than its particle phase concentrations which would support this potential artefact. Contamination of the PUF can happen not only inside the air sampler but can also happen in the lab due to micro-abrasion of material present in the lab as well. Have the authors randomly "prove" the precleaned PUFs before deployment to see if they were really "clean" by re-extracting the precleaned PUFs and analyzing the second extract? In any case, I would suggest the authors discuss breakthrough and general sampling artefacts and

contamination separately in section 3.1 rather than lumping all the reasons together to explain the observed "breakthrough" of BDE 183 and 209.

The analysis of the relationship between air concentrations (g, p, g+p) and meteorological conditions:

Table S5: Pearson correlation (linear relationship) analysis is used here instead of Spearman correlation (monotonic relationship). Is there any reason why Cg or Cp or Ctot would be linearly correlated with 1/T or any other met parameters? Thermodynamically speaking, there should be a linear relationship between natural-log transformed Cg (ln Cg) and 1/T (the Clausius-Clapeyron relationship), which provides information on the relative importance of volatilization from local sources and LRAT (Wania et al. ES&T, 1998, 32: 1013-1021), not Cg and 1/T. This relationship is explored in Figure S3. It is puzzling to try correlating C with 1/T in Table S5. If one only wants to know if C increases or decreases when T or any meteorological parameter increases, then a Spearman correlation should be used here. p. 8 line 29 "...gaseous concentration of all PBDEs...increased with ambient temperature (Table S5, Figure S3)". Figure S3 shows ln C versus 1/T while Table S5 shows C versus 1/T. This is very confusing. I suggest removing the correlation of C with 1/T in this table and focus the discussion on Figure S3 which would also tell the readers how C varies with temperatures. P. 9 Line 3-5 The authors suggests that there are other processes which controls Cg other than air-surface exchange. The authors should also refer to Wania et al. (1998) and point out that the shallow slopes for BDE 28 and 66 between ln Cg and 1/T suggest influence from LRAT which is a good reason for these lighter PBDEs which are relatively more volatile. Please show the p-values for the regressions in Figure S3.

As Referee #1 already pointed out, BDE 66 shows higher concentration for lower temperatures not vice versa. Also it says ABL height (shown as hmix in Table S5) shows strong correlations with Cp except BDE 28 on p. 8 line 8, but in Table S5, it seems that only BDE 85 didn't show a significant relationship, not BDE 28. Please correct.

[Figure]

Application and discussions of g-p partitioning models:

Was the fOM used in the equations the average value of PM10 concentrations (as the caption of Fig. S8 and S9 suggested) or the actual PM10 concentrations measured during each week of sampling? I presume that there is continuous measurement of PM10 at Košetice observatory? If the overall average value for the 4 years was used, please explain why you have not used the corresponding weekly average PM10 which I would suspect to vary quite a lot in different seasons, as well as over the years.

I am surprised that the authors have not pointed out the potential that the interference term from wet and dry deposition in the steady state model may be site specific and depends on the properties of the particles (including size distribution and physical composition). This would have partially explained why measured PBDE g-p partitioning are contradictory at a global scale which they have pointed out on p. 9.

Why didn't the authors try to use the ppLFER-type models proposed by Arp et al. (2008) and Shahpoury et al. (2016) to see if they give a better description of the g-p partitioning observed here? These models also take into account the makeup of the particles which may perform better than the 3 models used here that only consider the phys-chem properties of PBDEs.

p. 11 line 15 Should this sentence read "...6 and/or 6' position, such as for BDE 154" here? For BDE 154 (2,2',4,4',5,6'-BDE), there is only one Br at the 6' position. In the text, it says that the effect of stronger adsorption for the planar structure of BDE 153 as compared to BDE 154 which has a twisted structure is observed in the air monitoring results. If I am reading Figure S10 correctly, it seems that the measured particulate fraction of BDE 153 were lower than that of BDE 154 (e.g. a $\theta$BDE153 = 0.6 corresponds to a $\theta$BDE154 of 0.6-0.8). This means there is more BDE 154 sorbed to particles than BDE 153 which is opposite to what is stated in the text. Also, it seems that BDE 47 (2,2',4,4'-BDE) and 66 (2,3',4,4'-BDE) shows similar relationship in Figure S10 although none has a Br in the 6 or 6' position. Is there any explanation of this

relationship?

Trend analysis:

As Referee #1 pointed out, there is no analysis on seasonal variations at all. The authors should take the opportunity to analyze for temporal trends to better understand the seasonal variations in concentrations. Why only used a first-order relationship to try to develop time trends? Venier et al. (2012, ES&T, 46: 3928-34) compared 4 methods for deriving time trends for POPs. The authors can consider using any of the 4 methods, which take into consideration seasonal variations, to derive time trends.

The comparison of trends from literature can be updated with new trend information from the Great Lakes reported in Shunthirasingham et al. (2018, ESPI, 20: 469-479).

The figures in the main article look blurry, please re-make them.

Referee #1 noticed that there are missing references in the list. Also, it seems that some references are not typed in correctly, e.g. Davie-Martin et al. (2016) is missing a co-author's name. Please carefully check all references.

Minor: p. 2 line 7 . . .once PBDEs enter the air, they would partition between. . . p. 6 line 1 Suggest to remove the word "Indeed" which is a strange connector for these two sentences. p. 14 line 15, . . .the minimal criterion. . .

---

## Referee Comment (RC3) · Anonymous Referee #2 · 25 Apr 2018

In this manuscript, the authors present an analysis of PBDEs atmospheric concentrations for samples collected at a background station in Czech Republic over a 4 years period. The authors analyzed seasonality in the data as well as gas-particle partitioning. The dataset is interesting and they can provide some useful insights into the atmospheric concentrations of PBDEs in Europe. The manuscript though needs some work before it can be published.

General comments: QA/QC: I have some concerns regarding the data that the authors didn't address at all. Samples from 2011-12 were extracted and cleaned using

a method significantly different from those from 2013-2014. Also, samples from two different subsets (2011 and 2011-2014) were analyzed using two different instruments, columns and conditions. When datasets are analyzed using different methods, the issue of consistency and comparability needs to be addressed and this is especially important for long term data series. This comments dribbles down also to other QA/QC parameters such as blanks, and limit of detection /quantitation. It's not clear how this issue was dealt with for blanks: how were blanks calculated (e.g. annually or over the 4 years)? It's generally preferred to do it annually since it reflects more accurately lab practices at the time of processing. This dataset is very valuable and provides useful information for scientists and legislators but at the moment it is tainted by this QA/QC problem. The authors need to demonstrate that there is comparability and that their results are not affected by analytical issues. Breakthrough: Given the extremely large volumes collects, I am surprised that the breakthrough is so limited. Nevertheless, the breakthrough for BDE209 and BDE183 is a bit unsettling. I agree with the other reviewer in that it's particularly interesting that in certain samples 100% of these two congeners were detected in the second PUF. The authors speculate that this effect could be due to lab contamination but lab blanks would clearly reflect that and blank subtraction would equalize samples. A relatively simpler explanation that the authors didn't consider in the paper is the filter pore size. Here the filter cutoff is 2.2 um, which is quite high. For example, IADN employs QFF with a cutoff of 0.3 um. It's quite plausible that fine particles slips through the filter and end us in the PUF. This behavior should also be taken into account for the gas-particle partitioning. Factors affecting inert sample variations: Seasonality was not discussed or introduced before. As reviewer 1 noted, here seasonality is confused with ambient temperature, which is a cause but not an effect. Seasonality should be treated separately from the analysis with met data. The authors can not draw any conclusions on seasonality just based on the 1/T analysis (see page 8 lines 17 and 33, for example) The lack of relationship with most of meteorological parameters excluding temperature, is not surprising nor specific to PBDEs. Hafner and Hites showed that directional terms did not generally improve the

regression models (Environ. Sci. Technol. 39, 20, 7817-7825) for most SOCs. The results of the Pearson correlation analysis reported in Table S5 are so scattered that I find hard to draw any solid conclusion on these relationships. For example, why would BDE47 have a negative significant correlation with $1/T$ and BDE 66 a negative one? Gas-particle partitioning and modeling: the measured values for the particle fractions are certainly affected by the large filter cutoff, as discussed above. This artifact is certainly playing a significant role in the modeling and consequent interpretation. It is quite clear that the Koa model does a better job at describing this relationship than the other ones. If the gas phase concentrations were overestimated based on the larger than usual cutoff of the filters, the Kp would be smaller than expected. In this scenario, rather than the Koa based model overestimating the Kp, it's the measured Kp that is underestimated. I find that excluding BDE209 from the modeling is introducing a bias in the analysis and results. The authors should at least clarify why they chose to exclude it. Inter-annual variations: Seasonality is generally quite strong and its effect should be removed when calculating halving times. As mentioned by reviewer 3, there are a number of regression models that take into account seasonality than can be employed here.

Specific Comments: Page 3 Why is the use of the PM 10 separator never discussed in the manuscript other than at line 6 here? Perhaps I am missing something. Page 3 Bottom half Remove references to PCBs and dioxins since they are not relevant here. Pages 1-2 The use of term novel here is out of place, I am afraid. The authors didn't clarify what is the novel aspect of this study. Page 6, Line 16 How was the 4% under-estimation calculated? Page 6, Line 9 The reference to indoor studies is unnecessary since it's unfair to compare the two concentrations. Page 7, line 16 Please use more up to date reference for North America (see Liu et al., / Environment International 92–93 (2016) 442–449 and Ma et al., 2013). Page 7, line 26 Table S2 I wonder if this volume of 5264 m3 is a representative number. In line 11, the authors report that the sampling volume ranged from 4015 m3 to 5864 m3 for samples collected in 2015. The average is closer to 5000 m3. Page 9, line 6 Backward air trajectory was not properly

introduced and it seems abruptly introduced here. Page 13, line 11 Add also Liu et al., 2016. Page 13, lines 20-1 What was n in this partial regression? How was autumn and summer defined? I am quite wary of results involving BDE66 as mentioned above. Figures in main text: They are quite blurry and hard to read. Figure 2 Define the blue lines in caption. Figure 3 If trends are significant, include R and p value on plot. If they are not significant, remove the trend line. Table S4 I am quite surprised about BDE-66 levels. This congener is generally not that abundant in air and it wasn't a major one in commercial formulations. Since it elutes in a region that is quite crowded, I wonder if the peak was mistaken for something else. My hypothesis is reinforced by other places where BDE66 behaves differently than similar congeners (e.g BDE47); for example, in Table S3, the breakthrough behavior of BDE66 is remarkably different from that of BDE47, although admittedly this might have something to do with detection limits. Table S8 There is a more recent paper on temporal trends for samples around the Great Lakes (see Liu et al., / Environment International 92–93 (2016) 442–449) where data for 2005-2013 were used. Figure S12 If trends are significant, include R and p value on plot. If they are not significant, remove the trend line.

---

## Author Comment (AC1) · 21 Jun 2018

We would like to thank the reviewers for their thoughtful reading, comments and questions, which considerably helped to improve this manuscript. We have addressed all comments below and have indicated the corresponding modifications in the revised version of the manuscript.

Referre #1:

MS No.: acp-2018-144 Title: Are atmospheric PBDE levels declining in Central Europe? Examination of the seasonal variations, gas-particle partitioning and implications for long-range atmospheric transport Author(s): Céline Degrendele et al. MSType: Research article
General comments The manuscript presents four-year monitoring data (2011-2014) on atmospheric polybrominated diphenyl ethers (PBDEs) at the Košetice observatory, in an agricultural region in central Czech Republic. Particle- and gas-phase samples were collected on a weekly basis (7-day sampling duration) using a high-volume air sampler with PM10 pre-separator equipped with QFFs and 2 PUF plugs in series.
PBDEs analysis was performed on 101 valid weekly samples (31 from 2011, 25 from 2012, 20 from 2013, 25 from 2014). Correlations were examined between the individual concentrations (g, p, g+p) of PBDEs and prevailing meteorological parameters. The g/p partitioning of PBDEs, with the exception of BDE209, was investigated by comparing experimental partition coefficient, Kp, values with those determined by three predictive models, the KOA model, a steady state model proposed in literature, and a regression model based on the quantitative structure-property relationship (QSPR) also proposed by other investigators. The apparent halving times (_ 1/2) were calculated to investigate potential declining trends. The LRAT was also assessed by evaluating the backward trajectories of a small number of samples (10) using the Lagrangian particle dispersion model FLEXPART.
The authors have used appropriate methods for sampling/analysis of PBDEs and a thorough QA-QC procedure. The manuscript contains interesting data concerning the g/p partitioning behavior of PBDEs and the semi-longterm trends of their atmospheric levels at a background area of Central Europe.

My major concern is the large and variable amounts found for many PBDEs in the 2nd PUF plug. Since these amounts were included in the gas phase fraction probably resulted to underestimation of the particle fraction _measured.
The breakthrough analysis we had applied was based on raw data, which were not field blank corrected (this was done on the sum of the PUFs after the breakthrough analysis). This was inappropriate and we apologize for this mistake. Upon appropriate field blank correction, we obtain similar results for most congeners, but with much lower detection frequencies in the downstream (second) PUF. The interpretation of the highest masses of BDE183 and BDE209 on the lower PUFs have been updated taking into account the considerations of all reviewers (See updated Section 3.1). As mentioned in the manuscript, we consider the sampling configuration to be adequate to trap efficiently all PBDEs in the gaseous phase except BDE209.

Another question is why the subcooled-liquid–vapor pressure (PL)-based model was excluded from the g/p partitioning analysis.
We refrained from exploring $\log K_P = f(\log p_L)$ as the temperature dependence of vapour pressure is also reflected in the $\log K_P = f(\log K_{oa})$ plots (see e.g. Pankow and Bidleman, 1992; Cetin and Odabasi 2008; Lammel et al., 2010). Previously, it was common to test another vapour pressure based model i.e., the Junge-Pankow adsorption model (Pankow 1987). Such a model, implicitly assuming that adsorption is dominating gas-particle partitioning of the substances under study, is generally not promising for hydrophobic substances, which gasparticle partitioning is expected to be dominated by absorption in particulate organic matter (Finizio et al., 1997; Lohmann and Lammel, 2004; Goss and Schwarzenbach, 2001). The Junge-Pankow model has nevertheless been tested for PBDEs (Chen et al., 2006) including on another set of aerosol samples we collected and analysed (Besis et al., 2017). These results had confirmed the deficiency of this model and the perception that adsorption is not a significant process for PBDE gas-particle partitioning. Therefore, we prefer to not include this model in the discussion on gas-particle partitioning.

The exclusion of BDE209 from all g/p partitioning models needs explanation.
Two of the presented models used KOA as one of the critical parameter. To the best of our knowledge, given the analytical issues with BDE209, there are no measured KOA as a function of temperature for this compound available. For all remaining BDEs, we have used measured KOA relationships. It is therefore evident that an estimation of KOA as a function of the temperature will be associated with higher uncertainties than the measured values. Moreover, there are higher uncertainties with the reported measured particulate fraction for BDE209, we therefore prefered to exclude this compound from the G/P modelling.
*The manuscript now includes (at the beginning of the section on G/P modelling): "BDE209 was not considered in the different modelling approaches for two main reasons. Firstly, higher uncertainties are associated with the measured particulate fractions for this compound (see Section 3.1). Secondly, two of the tested models are based on $K_{OA}$ and the temperature dependence of this parameter is not available (never determined). "*

In addition to the above deficiencies, the manuscript needs substantial revision concerning various obscurities, inconsistencies, lacking information data, and missing references in the reference list. Finally, there is much room for language use improvement.
All specific comments have been answered consequently and the corresponding parts of the manuscript were modified.

Specific comments Title: The manuscript does not provide information on the seasonal variations of PBDEs levels, therefore "seasonal variations" in the title shall be replaced by "semi-longterm variations".
We have now included the analysis of seasonal variations in the manuscript and have also added semi-long term variations to the title.

2.2 Sample preparation and analysis It is obscure here whether the authors used a different preparation procedure for samples collected in 2013 and 2014 than those used for samples from 2011 – 2012. Please, clarify.
We are sorry about the confusion. Indeed, a different procedure was used for samples collected prior 2013 and those collected after.
*The manuscript now includes : ""The clean up and fractionation method differed between samples collected prior and those after 2013"*

2.3 Modelling of gas-particle partitioning
P.5. L. 27: the measured fOM value for this site shall be provided.
We have used $f_{OM}$ values provided by the Czech Hydrometeorological Institute which were measured every sixth day at the sampling site.
*The manuscript now includes: "The $f_{OM}$ were derived from the atmospheric concentrations of organic carbon (a conversion factor of 1.8 was used) which was determined every sixth day and were ranging from 0.07 to 0.98 with an average value of 0.39 ± 0.19."*

P.5. L. 29-30: The statement "Given the uncertainties while estimating such important physicochemical properties as KOA from other parameters, we did not consider BDE209 in the predictions" needs further clarification. Do the authors mean that the uncertainties for estimating KOA from other parameters is larger for BDE209 than for the lower PBDE congeners? In any case, the exclusion of BDE209 from all g/p partitioning models shall be explained.
See previous answer (exclusion of BDE209)

3.1 Breakthrough and sampling artefacts In the breakthrough experiments on the 25 samples from 2012, a significant amount of PBDEs was found in the lower PUF plug, particularly for BDE183 and BDE209 (on average 31.9% and 53.6 % of their total gasphase concentrations, respectively). ă˘A ´c Possible contamination of the PUFs from the electronic/plastic parts of the air sampler is considered as one of the reasons. However, in such a case, the contamination level would be the same in each sampling. Did the authors check that?
See previous answer (amounts in the 2nd PUF).

ă˘A ´c The authors say that these findings are in agreement with a detailed breakthrough study previously published (Melymuk et al., 2016a), however this reference is not in the reference list. Also, in their explanation for possible volatilisation loss from the filter, they cite Melymuk et al., 2014, which is also missing from the reference list.
We apologize for that. The reference list has been corrected and updated.

3.2 PBDE concentration levels
P.7, L. 5: The average gasand particle-phase concentrations of BDE209 provided in Table S5 (0.513 and 0.257 pg m-3, respectively) seem to be in discrepancy with the average measured particulate fraction (_measured) presented in Figure S6, which ranges between 55-85% in the four seasons. Please, check and correct if needed.
Indeed, these two datasets are in discrepancy, but correct. The average gaseous concentration of BDE209 was biased by few outliers (characterised by the high SD). The seasonal mean particulate mass fraction (FigureS3) was derived from the particulate mass fractions of individual samples. No changes made.

P.7, L. 7: Degrendele et al., 2016 cited here is missing from the reference list. Please, provide it.
We apologize about that. Now added.

Besis et al., 2017 could also be cited at this point as providing PBDEs concentrations at background sites in Europe.
This has now been added.

ă˘A ´c P.7, L. 10: Melymuk et al., 2016b is not in the reference list.
We do not anymore cite this article.

ă˘A ´c P.7, L. 9, 15, 16, 18: Besis and Samara, 2012 is not in the reference list. Actually, Besis and Samara 2012 is not dealing with the g/p partitioning of PBDEs. Perhaps the authors wanted to cite Besis et al., 2016 (Atmospheric occurrence and gas-particle partitioning of PBDEs at industrial, urban and suburban sites of Thessaloniki, northern Greece: Implications for human health, Envir. Poll. 215 (2016) 113-124).

Actually this section is not dealing with gas-particle partitioning and we consider that the information reviewed by Besis and Samara (2012) is relevant to support the points made with regard to the congener profiles. No changes made.

3.3 Factors affecting the inter sample variations

ăˇA ´c P. 8, L.10-12: The statement "In this study, when considering the total concentrations of individual PBDEs, a significant influence of ambient temperature 10 was suggested only for BDE47 and BDE66 (higher concentrations for higher temperatures) and BDE153, BDE154 and BDE183 (higher concentrations for lower temperatures) (Table S5)"is not true! Table S5 shows negative correlation with 1/T (i.e. positive with T) only for BDE47, while positive for BDEs 66, 153, 154, 183. Please, correct properly.

We apologize for that error and have now corrected. Moreover, BDE66 has now been removed from this manuscript (see following comment by Reviewer 2).

ăˇA ´c P.8. L.21: p>0.05 shall be p<0.05 here.

Now corrected.

ăˇA ´c Seasonality is confused here with the correlation with ambient T. Unfortunately, seasonal variations of PBDEs levels are not examined in the manuscript. Correlations with ambient T are as expected. Why the authors did not provide Clausius-Clapeyron plots for the gas-phase concentrations?

The investigation of seasonality on PBDEs atmospheric concentrations is now included in the Section 3.6.

An investigation of Clausius-Clapeyron equation is now included. However, given the important seasonality in the measured particulate fraction of most PBDEs investigated, we do not consider that this is relevant.

*The manuscript now includes:"An examination of the temperature dependence of the PBDEs gaseous concentrations using the Clausius-Clapeyron equation (see Supplement) was done and results are presented in Table S10. Significant correlations were found between the natural logarithm of partial pressure versus the inverse of ambient temperature for all PBDEs, except BDE28 and BDE209. This suggests that the gas-phase concentrations of these two congeners are not controlled by temperature dependent sources. This lack of temperature dependence has been previously attributed to long-range atmospheric transport (Hoff et al., 1998; Wania and Haugen, 1998). However, at least for BDE28, we suggest that the photolytic debromination of higher brominated congeners (Bezares-Cruz et al., 2004; Wei et al., 2013) may also play a role. In case of the remaining congeners, the strong influence of ambient temperature on the gaseous concentrations of PBDEs, characterized by the high slopes in Table S10, has been often interpreted by previous studies (Cetin and Odabasi, 2008; Davie-Martin et al., 2016) as a demonstration that PBDE gaseous concentrations are controlled by revolatilisation from surfaces (soils or waters). However, given the large influence of ambient temperature on $\theta_{measured}$ (see Section 3.4), it is uncertain that the gas-phase concentrations of PBDEs are controlled by air-surface exchange rather than by revolatilisation from the particles. Therefore, we would suggest to focus the interpretation of Clausius Clapeyron equation only for those substances which are mainly in the gas-phase (i.e. $\theta_{measured} < 0.2$), regardless of the ambient temperature. "*

ăˇA ´c The statement in P.8. L. 17-18 "Overall, the absence of seasonality in the total concentrations of most PBDEs suggests that their atmospheric levels are still driven by primary sources." shall be "Overall, the absence of correlation of the total concentrations of most PBDEs with ambient temperature suggests that their atmospheric levels are still driven by primary sources."

This statement is not anymore included in the manuscript.

3.2 Gas-particle partitioning in air samples

ă˘A ´c P.9. L. 28: Again seasonality is confused with the correlation with ambient T. Please, correct properly.

Changed accordingly.

ă˘A ´c P.9. L. 30: The finding that the temperature is an important variable affecting the partitioning of PBDEs in the atmosphere is not new, it has been shown in all similar studies.

This sentence now removed.

The authors could provide the logKp-T relationship as well in addition to the correlation coefficient between _measured and 1/T.

This is now included (Table S5).

ă˘A ´c P.12:

I think that the first reason for non considering that the deviation from equilibrium was due to the influence of wet and dry deposition should be the comparison of the particle fraction of PBDEs between samples with high and low precipitation height. I suggest changing the order of reasons.

Thank you for the suggestion. We have now changed the order of reasons.

3.5 Modelling of gas-particle partitioning ă˘A ´c P.10. L. 12: Please change "seasonal" to "temporal".

Changed accordingly.

ă˘A´c It would be interesting if the authors showed and discussed the logKp-logPLΣ relationship.

Please, see reply above: we refrain from testing gas-particle partitioning of PBDEs on adsorption, as the process is rather determined by absorption.

3.2 Inter-annual variations

ă˘A ´c The statement "$C_{0,i}$ is the theoretical concentrations of individual PBDE measured at $t_0$ (i.e. the end date of PBDE production)" is not clear. The authors have to further explain if and how they estimated the $lnC_{0,i}$ data used in Eq. (8) and if these data are representative for Central Europe.

We have now changed the regression model applied to our dataset, and this term is not included anymore.

ă˘A ´c Is it sure that T units in Eq. (8) are days and not years? Please, confirm.

Yes, T is in days, as we are now also investigating the seasonal variations.

ă˘A ´c It should be clarified that the total (g+p) concentrations were used for $C_i$ in Eq. (8). Why the apparent half-lives were not calculated separately for the two phases?

We have now indicated that total concentrations were used. A derivation of apparent half-lives for individual phases in atmospheric aerosols would be misleading: characteristic times of interphase conversions are much shorter. Hence, there is no ,live' in one of the phases on the time scale of the study.

Conclusions

Supplementary Material ă˘A ´c Table S3: For clarity reasons, please change "% of compound mass found on the lower PUF" to "% of gas-phase compound mass found on the lower PUF".

â˘A ´c Table S6: Please change title to: "Results of regression analysis between _measured and the inverse of temperature (K-1) for individual congeners. Numbers in bold indicate cases for which regression coefficients (r2) were statistically significant (p<0.05)" â˘A ´c Table S7: Please change title to: "Results of Pearson correlation analysis between _measured and the precipitation rate for individual PBDEs. Numbers in bold indicate cases for which the correlations were statistically significant (p<0.05)". â˘A ´c Table S8: Please change title to "Apparent half lives (_ ) of individual PBDEs observed in this study and elsewhere: : :..Â˙z â˘A ´c Figure S3: Please change legend to: "Correlation between the gaseous concentration of individual PBDEs (ln transformed) with the inverse of temperature".

All of these changes now included in the manuscript and Supplement.

Referre #3

This paper presents 4 years of air monitoring data of PBDEs (2011-2014) measured at the background site of Košetice observatory in Central Europe. Gas and particle phases of the air samples taken with a high volume air sampler equipped with a PM10 size-exclusion inlet were analyzed separately. The relationship between meteorological conditions and PBDE air concentrations in the particle phase, in the gas phase and total g+p concentrations were examined. Gas-particle partitioning coefficients (Kp) estimated by three models, namely KOA-based model, steady-state model and a quantitative structure-property relationship (QSPR) model proposed by Wei et al. (2017) were compared with observed Kp. It was found that none of the models provide satisfactory prediction of the gas-particle partitioning observed for PBDEs measured in ambient air and the authors tried to explain why this is the case. Back trajectories were used to examine potential sources of PBDEs in samples with the highest and the lowest PBDE concentrations. Temporal trends of PBDEs measured were assessed using a simple regression method to estimate the first-order halflives which suggest declining trends. The authors suggest that after PBDEs have been regulated under the Stockholm Convention globally, debromination from higher BDE congeners may result in the enrichment of lower brominated congeners which are more persistent and are more mobile than heavier congeners. This manuscript presents an interesting and valuable air monitoring dataset of PBDE measured in Central Europe. The authors did a fairly thorough analysis of the gasparticle partitioning observed and relationships of air concentrations with meteorological conditions. I found the fact that none of the three theoretical models provide satisfactory g-p partitioning observed interesting and the authors' attempt to explain why this is the case helpful. However, there are a few issues which I'd like to raise to help improve the manuscript and they are given below.

QA/QC:
Blank correction: Were the sample blank corrected using the annual average of the field blanks or the average of all 4 years of blanks? It is recommendable that the samples be blank corrected with the annual average field blank for 2 reasons: 1. The background levels for PBDEs in the lab may vary over time depending on what was being used and exists in the lab (e.g. old cardboard boxes containing PBDEs etc.); and 2. as this is a long-term air monitoring site, it would be problematic in the future if the samples were not blank corrected using annual average field blank, i.e. after collection of a few more years of samples, the LOQs would change. Consistent data management over the long term is important in generating a consistent dataset for the determination of temporal trends of POPs.

Indeed, the samples were blank corrected using the annual average of field blanks and not the average of all 4 years of blanks.

*The manuscript now includes: " The PBDE concentrations presented here were blank corrected by subtracting the average of the field blanks on an annual basis, separately for GFFs and PUFs."*

Breakthrough and sampling artefacts:

I am in fact not very concern about breakthrough in the PUF resulting in underestimation of the gas phase concentration. In Bidleman and Tysklind (2018, Chemosphere 192: 267-271), it was demonstrated that when PUF2/PUF1_0.5, the collected fraction exceeds 90 %. Given the case that <20% of the lower congeners were found on PUF2, most of the PBDEs were probably adequately captured by the PUF1+PUF2 sampling train. I'm more concern about the fact that the maximum percentage of BDE 183 and 209 found on PUF2 was 100%, meaning that nothing was found on PUF1 for some samples. It is impossible that there was 100% breakthrough. This looks more like contamination than breakthrough, especially with BDE 209 which usually has high background levels. P. 6 Line 16, how was this underestimation of "up to 4%" determined? For the reasons given for the "breakthrough" of BDE 183 and 209, the fact that there may be blowoff from the filter should affect all congeners rather than just BDE 183 and 209 alone. It is more a sampling artefact than an explanation for the "breakthrough" observed. If it was blowoff from the filter, one should see BDE 183 and 209 more in PUF1 than in PUF2, i.e. it cannot explain the up to 100 % mass found in PUF2. Also, Okonski et al. (2014) found most of the PBDEs on aerosols <0.95 um and the QFF has a pore size of 2.2 um, have the authors considered fine particles physically breaking through the QFF into the PUF below? With the long sampling duration of 7 days and high flow rate of 31.3 m3/h, physical breakthrough of fine particles is possible. Of course, again this does not explain the high percentage mass found on PUF2.

Looking at Table S4, when BDE 209 was detectable in gas phase (although it's not very often 41 %), it seems that its gas phase concentrations were higher than its particle phase concentrations which would support this potential artefact. Contamination of the PUF can happen not only inside the air sampler but can also happen in the lab due to micro-abrasion of material present in the lab as well. Have the authors randomly "prove" the precleaned PUFs before deployment to see if they were really "clean" by re-extracting the precleaned PUFs and analyzing the second extract? In any case, I would suggest the authors discuss breakthrough and general sampling artefacts and contamination separately in section 3.1 rather than lumping all the reasons together to explain the observed "breakthrough" of BDE 183 and 209.

We would like to thank the referee for her/his precious comments.

We need to apologize for two mistakes we did. Firstly, it was mentioned that the pore size of the filters used were 2.2 µm. However, this was not the pore size of the filter, which is not indicated by the manufacturer, but this was the cutoff for particle retention in liquid. The same filters are used by US EPA PM$_{10}$ Ambient Air Monitoring and do fulfil the relevant criteria.

Secondly, the breakthrough analysis was previously performed on the raw data. Then the PUFs were summed up, and then the annual mean of field blank concentrations was substracted, which was inappropriate. We have now updated the calculation by firstly performing field blank substraction prior the breakthrough analysis. As pointed out, many of the previously reported concentrations on PUFs were within the blank levels and the detection frequencies are now lower. Still, BDE183 and BDE209 are frequently positively found on the lower PUF. We have now updated this section of the manuscript taking into account all comments from reviewers and discuss breakthrough, sampling artefacts and possible contamination separately (see Section 3.1).

The analysis of the relationship between air concentrations (g, p, g+p) and meteorological conditions:

Table S5: Pearson correlation (linear relationship) analysis is used here instead of Spearman correlation (monotonic relationship).

Is there any reason why Cg or Cp or Ctot would be linearly correlated with 1/T or any other met parameters? Thermodynamically speaking, there should be a linear relationship between natural-log transformed Cg (ln Cg) and 1/T (the Clausius-Clapeyron relationship), which provides information on the relative importance of volatilization from local sources and LRAT (Wania et al. ES&T, 1998, 32: 1013-1021), not Cg and 1/T. This relationship is explored in Figure S3. It is puzzling to try correlating C with 1/T in Table S5. If one only wants to know if C increases or decreases when T or any meteorological parameter increases, then a Spearman correlation should be used here.

We apologize for that error. We have now updated Table S5 using a Spearman correlation. Moreover, we have now considered the relationship only between lnC and 1/T.

An investigation of Clausius-Clapeyron equation is now included. However, given the very strong influence of ambient temperature on the measured particulate mass fraction of most PBDE congeners, we do not consider that this is relevant.

*The manuscript now includes:"An examination of the temperature dependence of the PBDEs gaseous concentrations using the Clausius-Clapeyron equation (see Supplement) was done and results are presented in Table S10. Significant correlations were found between the natural logarithm of partial pressure versus the inverse of ambient temperature for all PBDEs, except BDE28 and BDE209. This suggests that the gas-phase concentrations of these two congeners are not controlled by temperature dependent sources. This lack of temperature dependence has been previously attributed to long-range atmospheric transport (Hoff et al., 1998; Wania and Haugen, 1998). However, at least for BDE28, we suggest that the photolytic debromination of higher brominated congeners (Bezares-Cruz et al., 2004; Wei et al., 2013) may also play a role. In case of the remaining congeners, the strong influence of ambient temperature on the gaseous concentrations of PBDEs, characterized by the high slopes in Table S10, has been often interpreted by previous studies (Cetin and Odabasi, 2008; Davie-Martin et al., 2016) as a demonstration that PBDE gaseous concentrations are controlled by revolatilisation from surfaces (soils or waters). However, given the large influence of ambient temperature on $\theta_{measured}$ (see Section 3.4), it is uncertain that the gas-phase concentrations of PBDEs are controlled by air-surface exchange rather than by revolatilisation from the particles. Therefore, we would suggest to focus the interpretation of Clausius Clapeyron equation only for those substances which are mainly in the gas-phase (i.e. $\theta_{measured} < 0.2$), regardless of the ambient temperature. "*

p. 8 line 29 ": : :gaseous concentration of all PBDEs: : :increased with ambient temperature (Table S5, Figure S3)". Figure S3 shows ln C versus 1/T while Table S5 shows C versus 1/T. This is very confusing. I suggest removing the correlation of C with 1/T in this table and focus the discussion on Figure S3 which would also tell the readers how C varies with temperatures.

We have now removed the correlation of C with 1/T and focused the discussion on lnC vs. 1/T.

P. 9 Line 3-5 The authors suggests that there are other processes which controls Cg other than air-surface exchange. The authors should also refer to Wania et al. (1998) and point out that the shallow slopes for BDE 28 and 66 between ln Cg and 1/T suggest influence from LRAT which is a good reason for these lighter PBDEs which are relatively more volatile.

Please, see the previous comment on Clausius Clapeyron plots.

Please show the p-values for the regressions in Figure S3.

Now indicated in the caption.

As Referee #1 already pointed out, BDE 66 shows higher concentration for lower temperatures

not vice versa. Also it says ABL height (shown as hmix in Table S5) shows strong correlations with Cp except BDE 28 on p. 8 line 8, but in Table S5, it seems that only BDE 85 didn't show a significant relationship, not BDE 28. Please correct.

We are sorry about these mistakes which have now been corrected. Please note that BDE66 was removed from this article.

Application and discussions of g-p partitioning models:

Was the fOM used in the equations the average value of PM10 concentrations (as the caption of Fig. S8 and S9 suggested) or the actual PM10 concentrations measured during each week of sampling? I presume that there is continuous measurement of PM10 at Košetice observatory? If the overall average value for the 4 years was used, please explain why you have not used the corresponding weekly average PM10 which I would suspect to vary quite a lot in different seasons, as well as over the years.

We have used fOM values provided by the Czech Hydrometeorological Institute which were measured every sixth day at the sampling site.

*The manuscript now includes: "The $f_{OM}$ were derived from the atmospheric concentrations of organic carbon (a conversion factor of 1.8 was used) which was determined every sixth day and were ranging from 0.07 to 0.98 with an average value of 0.39 ± 0.19."*

I am surprised that the authors have not pointed out the potential that the interference term from wet and dry deposition in the steady state model may be site specific and depends on the properties of the particles (including size distribution and physical composition). This would have partially explained why measured PBDE g-p partitioning are contradictory at a global scale which they have pointed out on p. 9.

Thank you for the suggestion.

*The manuscript now includes: "However, the term describing this influence in Eq 4 does not consider important characteristics of the site such as meteorological conditions (e.g. precipitation rate, temperature) or aerosol properties (e.g. mass size distribution, PM composition)."*

Why didn't the authors try to use the ppLFER-type models proposed by Arp et al. (2008) and Shahpoury et al. (2016) to see if they give a better description of the g-p partitioning observed here? These models also take into account the makeup of the particles which may perform better than the 3 models used here that only consider the phys-chem properties of PBDEs.

We had apply a pp-LFER model on our data, and the predicted $K_p$ was higher than the one determined by the $K_{OA}$ model. This is, actually, expected as more attractive molecular interactions beyond absorption in organic matter were considered. However, given the uncertainties related to input parameters and the preliminary nature of the model setup, we decided to not include this in the current manuscript. The application of ppLFER for interphase partitioning of PBDE in the atmospheric environment, including gas-particle partitioning is an on-going research of our group (Shahpoury et al; in prep).

p. 11 line 15 Should this sentence read ": : :6 and/or 6' position, such as for BDE 154" here? For BDE 154 (2,2',4,4',5,6'-BDE), there is only one Br at the 6' position. In the text, it says that the effect of stronger adsorption for the planar structure of BDE 153 as compared to BDE 154 which has a twisted structure is observed in the air monitoring results. If I am reading Figure S10 correctly, it seems that the measured particulate fraction of BDE 153 were lower than that of BDE 154 (e.g. a _BDE153 = 0.6 corresponds to a _BDE154 of 0.6-0.8). This means there is more BDE 154 sorbed to particles than BDE 153 which is opposite to what is stated in the text. Also, it seems that BDE 47 (2,2',4,4'-BDE) and 66 (2,3',4,4'-BDE) shows similar relationship in Figure S10 although none has a Br in the 6 or 6' position. Is there any explanation of this relationship?

*We thank the reviewer for noticing the mistake in Figure S10; the axes had been mislabelled and have now been corrected.*

*We had wanted to emphasize the importance of the Br atom in the ortho position (relative to oxygen substituted carbon atom). Not only the 6 and 6' positions fulfill this requirement, but also the 2 and 2' positions. In the case of BDE 154 (2,2',4,4',5,6'-BDE) there are three Br atoms in ortho positions. BDE 47 (2,2',4,4'-BDE) contains two Br atoms in ortho positions, whereas BDE 66 (2,3',4,4'-BDE) only contains one.*

*We acknowledge that using just the positional labels 6 and 6' could be confusing and unclear for the reader. To address this, we have updated the manuscript to discuss this concept in terms of ortho substituted Br (which should imply the 2, 2', 6 and 6' positions).*

*The manuscript now includes: "We highlight a study by Ding et al., (2014) which investigated the adsorption of different congeners on graphene (a structure that on a molecular level has similarities to black carbon). It was found that, in addition to the number of bromine atoms, the adsorption energy was also affected by the 3-dimensional structure of the PBDE congener. Specifically, steric interactions between bromine atoms in the ortho position (relative to the oxygen substituted carbon atom) appear to be important. This effect is best illustrated by congeners BDE153 and BDE154, both of which have the same number of bromine atoms. However, BDE154 has three Br atoms in the ortho position; this meant the congener adopted a twisted structure and adsorbed more weakly onto the graphene surface. BDE153 on the other hand, with only two Br atoms in the ortho position, can adopt a planar structure and adsorb more strongly. The consequences of this effect are observed in our results (Figure S10). We also note similar behaviour between BDE99 and BDE100; it appears that congeners with more Br atoms in the ortho position tend to have smaller particulate fractions when compared with other congeners of the same mass. Such effects are not captured by using KOA alone as a predictor (octanol having more degrees of freedom can better accommodate to twisted structures). We speculate these effects could influence the ability of specific BDEs to both adsorb onto and diffuse within the bulk condensed phases of PM."*

Trend analysis:
As Referee #1 pointed out, there is no analysis on seasonal variations at all. The authors should take the opportunity to analyze for temporal trends to better understand the seasonal variations in concentrations. Why only used a first-order relationship to try to develop time trends? Venier et al. (2012, ES&T, 46: 3928-34) compared 4 methods for deriving time trends for POPs. The authors can consider using any of the 4 methods, which take into consideration seasonal variations, to derive time trends.

Thank you for the suggestion. We have now applied one of the method used by Venier et al (2012), which also addresses seasonal variations. See Section 3.5

The comparison of trends from literature can be updated with new trend information from the Great Lakes reported in Shunthirasingham et al. (2018, ESPI, 20: 469-479).

Thank you for informing us about this interesting article. We have now included these results in the discussion on PBDE long term trends worldwide, as well as in the Table comparing the half-lives of PBDEs.

*The manuscript now includes:" Similarly, at two sites around the Canadian Great Lakes, PBDE concentrations were found to decrease slowly, with half lives in the range of 2-16 years and faster decline rates at the site closest to urban areas (Shunthirasingham et al., 2018)."*

The figures in the main article look blurry, please re-make them.

We have now improved the layout of all figures present in the manuscript.

Referee #1 noticed that there are missing references in the list. Also, it seems that some references are not typed in correctly, e.g. Davie-Martin et al. (2016) is missing a co-author's name. Please carefully check all references.

We have now carefully checked all references.

Minor: p. 2 line 7 : : :once PBDEs enter the air, they would partition between: : : p. 6 line 1 Suggest to remove the word "Indeed" which is a strange connector for these two sentences. p. 14 line 15, : : :the minimal criterion: : :

Thank you, modified accordingly.

Referee #2:

In this manuscript, the authors present an analysis of PBDEs atmospheric concentrations for samples collected at a background station in Czech Republic over a 4 years period. The authors analyzed seasonality in the data as well as gas-particle partitioning. The dataset is interesting and they can provide some useful insights into the atmospheric concentrations of PBDEs in Europe. The manuscript though needs some work before it can be published.

General comments:
QA/QC: I have some concerns regarding the data that the authors didn't address at all. Samples from 2011-12 were extracted and cleaned using a method significantly different from those from 2013-2014. Also, samples from two different subsets (2011 and 2011-2014) were analyzed using two different instruments, columns and conditions. When datasets are analyzed using different methods, the issue of consistency and comparability needs to be addressed and this is especially important for long term data series. This comments dribbles down also to other QA/QC parameters such as blanks, and limit of detection /quantitation. It's not clear how this issue was dealt with for blanks: how were blanks calculated (e.g. annually or over the 4 years)? It's generally preferred to do it annually since it reflects more accurately lab practices at the time of processing. This dataset is very valuable and provides useful information for scientists and legislators but at the moment it is tainted by this QA/QC problem. The authors need to demonstrate that there is comparability and that their results are not affected by analytical issues.

We apologize for not providing sufficient proof of the comparability between the results obtained from different sampling preparation and/or column. We have now added a table comparing the results of spiked PUFs from the two different sampling preparation methods as well as a table showing the changes of the relative response factors relqted to the different columns used.

*The manuscript now includes: "The different sample preparation and/or column used has a minor effect on the overall quality of the data (<12%, Tables S3 and S4). Therefore, the data obtained are directly comparable and suitable to derive long term trends"*

The samples were blank corrected using the field blanks generated in individual years, and not the average of the field blanks of 4 years.

*The manuscript now includes: " The PBDE concentrations presented here have been blank corrected by subtracting the average of the field blanks on an annual basis, separately for GFFs and PUFs."*

Breakthrough: Given the extremely large volumes collects, I am surprised that the breakthrough is so limited. Nevertheless, the breakthrough for BDE209 and BDE183 is a bit unsettling. I agree with the other reviewer in that it's particularly interesting that in certain samples 100% of these two congeners were detected in the second PUF. The authors speculate that this effect

could be due to lab  contamination but lab blanks would clearly reflect that and blank subtraction would equalize samples. A relatively simpler explanation that the authors didn't consider in the paper is the filter pore size. Here the filter cutoff is 2.2 um, which is quite high. For example, IADN employs QFF with a cutoff of 0.3 um. It's quite plausible that fine particles slips through the filter and end us in the PUF. This behavior should also be taken into account for the gas-particle partitioning.

We would like to thank the referee for her/his precious comments.

We need to apologize for two mistakes we did. Firstly, it was mentioned that the pore size of the filters used were 2.2 μm however, this was not the pore size of the filter, which is not indicated by the manufacturer, but this was the cutoff for particle retention in liquid. The same filters are used by the US EPA $PM_{10}$ Ambient Air Monitoring and do fulfil the relevant criteria. Secondly, the breakthrough analysis was previously performed on the raw data. Then the PUFs were summed up, and then the field blank substraction was performed, which was inappropriate. We have now updated the calculation by firstly performing field blank substraction prior the breakthrough analysis. As pointed out, many of the previously reported concentrations on PUFs were within the blank levels and the detection frequencies are now lower. Still, BDE183 and BDE209 are frequently positively found on the lower PUF. We have now updated this section taking into account all comments from reviewers.

Factors affecting inert sample variations: Seasonality was not discussed or introduced before. As reviewer 1 noted, here seasonality is confused with ambient temperature, which is a cause but not an effect. Seasonality should be treated separately from the analysis with met data. The authors can not draw any conclusions on seasonality just based on the 1/T analysis (see page 8 lines 17 and 33, for example) The lack of relationship with most of meteorological parameters excluding temperature, is not surprising nor specific to PBDEs. Hafner and Hites showed that directional terms did not generally improve the regression models (Environ. Sci. Technol. 39, 20, 7817-7825) for most SOCs. The results of the Pearson correlation analysis reported in Table S5 are so scattered that I find hard to draw any solid conclusion on these relationships. For example, why would BDE47 have a negative significant correlation with 1/T and BDE 66 a negative one?

We apologize for that confusion and have now included a discussion on seasonality in the part on semi-long term trends. We have now removed BDE66 from this manuscript and have shortened the discussion on the possible influence of meteorological parameters on the PBDE concentrations.

*The manuscript now includes:" No or low influence of wind speed and wind direction on the PBDE concentrations were observed, consistent with previous studies (Besis et al., 2015; Cetin and Odabasi, 2008), but also more generally consistent for POPs (Hafner and Hites, 2005)."*

Gas-particle partitioning and modeling: the measured values for the particle fractions are certainly affected by the large filter cutoff, as discussed above. This artifact is certainly playing a significant role in the modeling and consequent interpretation.

It is quite clear that the Koa model does a better job at describing this relationship than the other ones. If the gas phase concentrations were  overestimated based on the larger than usual cutoff of the filters, the Kp would be smaller than expected. In this scenario, rather than the Koa based model overestimating the Kp, it's the measured Kp that is underestimated.

The filtration efficiency, though not provided by the manufacturer, is very high even for nano-sized particles ('total filter', commonly used by monitoring networks, such as e.g., the US EPA). Therefore, we expect no significant influence of the filtration efficiency on measureed $K_p$'s.

I find that excluding BDE209 from the modeling is introducing a bias in the analysis and results. The authors should at least clarify why they chose to exclude it.

Two of the presented models used $K_{OA}$ as one of the critical parameter. To the best of our knowledge, given the analytical issues with BDE209, there are no measured $K_{OA}$ as a function of temperature for this compound available. For all remaining BDEs, we have used measured $K_{OA}$ relationships. It is therefore evident that an estimation of $K_{OA}$ as a function of the temperature will be associated with higher uncertainties than the measured values. Moreover, we should keep in mind that there are higher uncertainties associated with the reported measured particulate fractions. We therefore prefered to exclude this compound from this section.

*The manuscript now includes (at the beginning of the section on G/P modelling): "BDE209 was not considered in the different modelling approaches for two main reasons. Firstly, higher uncertainties are associated with the measured particulate fractions for this compound (see Section 3.1). Secondly, two of the tested models are based on $K_{OA}$ and the temperature dependence of this parameter is not available (never determined)."*

Inter-annual variations: Seasonality is generally quite strong and its effect should be removed when calculating halving times. As mentioned by reviewer 3, there are a number of regression models that take into account seasonality than can be employed here.

Thank you for the suggestion. We have now used a different regression model to quantify the semi long-term trends apart from the seasonality.

Specific Comments:
Page 3 Why is the use of the PM 10 separator never discussed in the manuscript other than at line 6 here? Perhaps I am missing something.

A defined cutoff is operationally preferable, rather than an undefined sampling of PM (limitations of isokinetic sampling etc.). $PM_{10}$ is the most common cutoff in air quality and aerosol research and exposure studies, as it addresses the inhalable size fraction. Moreover, PBDEs as most other SOCs are present mainly on fine particles (i.e. <1 µm, Okonski et al; 2014).

*The manuscript now reads: "The sampler addressed the inhalable size fraction, $PM_{10}$. PBDEs are mostly sorbed to fine and sub-micrometer sized particles (Okonski et al., 2014; Besis et al., 2017)".*

Page 3 Bottom half Remove references to PCBs and dioxins since they are not relevant here.
Changed accordingly.

Pages 1-2 The use of term novel here is out of place, I am afraid. The authors didn't clarify what is the novel aspect of this study.
We agree and have now removed the term "novel" from the manuscript.

Page 6, Line 16 How was the 4% underestimation calculated?
This was estimated by assuming that a third PUF would capture 20% of the second PUF and so on. However, we have now removed this estimation and included the reference from the more accurate estimation of Bidleman and Tysklind (2018).

*The manuscript now reads: " Given that Bidleman and Tysklind (2018) demonstrated that when less than 50% is found in the lower PUF plug, the collected gaseous mass fractions should be larger than 90%, we consider the current sampling configuration and sample preparation to be efficient for trapping all gas-phase PBDE congeners addressed, except BDE209."*

Page 6, Line 9 The reference to indoor studies is unnecessary since it's unfair to compare the two concentrations.
Removed reference to indoor air accordingly.

Page 7, line 16 Please use more up to date reference for North America (see Liu et al., / Environment International 92– 93 (2016) 442–449 and Ma et al., 2013).
We have now used only the latest available information from North America.

Page 7, line 26 Table S2 I wonder if this volume of 5264 m3 is a representative number. In line 11, the authors report that the sampling volume ranged from 4015 m3 to 5864 m3 for samples collected in 2015. The average is closer to 5000 m3.
The fictive volume of 5264 $m^3$ is a representative number as it is the average of all volumes for single samples (the median is 5344 $m^3$). Volumes lower than 5000 $m^3$ were rare.

Page 9, line 6 Backward air trajectory was not properly introduced and it seems abruptly introduced here.
We have now introduced it.
*The manuscript now reads: " LRAT is an important source of POPs such as PBDEs in background environments. The analysis of air mass history, as described in Section 2.5, was performed to identify potential source areas for PBDE in Central Europe."*

Page 13, line 11 Add also Liu et al., 2016.
Ok, done

Page 13, lines 20-1 What was n in this partial regression? How was autumn and summer defined? I am quite wary of results involving BDE66 as mentioned above.
We have now removed this part from the manuscript.

Figures in main text: They are quite blurry and hard to read.
We have now improved them.

Figure 2 Define the blue lines in caption.
Ok, done

Figure 3 If trends are significant, include R and p value on plot. If they are not significant, remove the trend line.
We have now updated accordingly this Figure.

Table S4 I am quite surprised about BDE-66 levels. This congener is generally not that abundant in air and it wasn't a major one in commercial formulations. Since it elutes in a region that is quite crowded, I wonder if the peak was mistaken for something else. My hypothesis is reinforced by other places where BDE66 behaves differently than similar congeners (e.g BDE47); for example, in Table S3, the breakthrough behavior of BDE66 is remarkably different from that of BDE47, although admittedly this might have something to do with detection limits.
We recognize that as no internal standard was available for this compound, there are higher uncertainties in congener identification. We have now removed this compound from the manuscript.

Table S8 There is a more recent paper on temporal trends for samples around the Great

Lakes (see Liu et al., / Environment International 92–93 (2016) 442–449) where data for 2005-2013 were used.

Thank you for letting us know about this interesting article. Previous data have now been replaced by these ones.

Figure S12 If trends are significant, include R and p value on plot. If they are not significant, remove the trend line.

We have now updated accordingly this Figure.

---

## Referee Report (RR1)

MS No.: acp-2018-144
Title: Are atmospheric PBDE levels declining in Central Europe? Examination of the seasonal variations, gas-particle partitioning and implications for long-range atmospheric transport
Author(s): Céline Degrendele et al.
MS Type: Research article

**General comments on the authors' response and the revised manuscript**
The revised manuscript is substantially improved, however needs further perfections before it can be accepted for publication in acp.

The authors' response (in green) was reasonable to most of my comments, although some issues remain unsolved (see below in red). In addition, there are several other issues raised that need to be addressed (see Additional comments on manuscript v.3).

**Comments on the original manuscript needing further attention**

Another question is why the subcooled-liquid–vapor pressure (PL)-based model was excluded from the g/p partitioning analysis.
We refrained from exploring $\log K_P = f(\log p_L)$ as the temperature dependence of vapour pressure is also reflected in the $\log K_p = f(\log K_{oa})$ plots (see e.g. Pankow and Bidleman, 1992; Cetin and Odabasi 2008; Lammel et al., 2010). Previously, it was common to test another vapour pressure based model i.e., the Junge-Pankow adsorption model (Pankow 1987). Such a model, implicitly assuming that adsorption is dominating gas-particle partitioning of the substances under study, is generally not promising for hydrophobic substances, which gas-particle partitioning is expected to be dominated by absorption in particulate organic matter (Finizio et al., 1997; Lohmann and Lammel, 2004; Goss and Schwarzenbach, 2001). The Junge-Pankow model has nevertheless been tested for PBDEs (Chen et al., 2006) including on another set of aerosol samples we collected and analysed (Besis et al., 2017). These results had confirmed the deficiency of this model and the perception that adsorption is not a significant process for PBDE gas-particle partitioning. Therefore, we prefer to not include this model in the discussion on gas-particle partitioning.

I agree that adsorption is less significant that absorption, nevertheless, the Junge-Pankow adsorption model was found to predict better than the KOA model the θmeasured in the warm season for the moderately brominated congeners DBE-49, -71, -47, and -66 (Besis et al., 2016).

The exclusion of BDE209 from all g/p partitioning models needs explanation.

Two of the presented models used KOA as one of the critical parameter. To the best of our knowledge, given the analytical issues with BDE209, there are no measured KOA as a function of temperature for this compound available. For all remaining BDEs, we have used measured KOA relationships. It is therefore evident that an estimation of KOA as a function of the temperature will be associated with higher uncertainties than the measured values. Moreover, there are higher uncertainties with the reported measured particulate fraction for BDE209, we therefore prefered to exclude this compound from the G/P modelling.

*The manuscript now includes (at the beginning of the section on G/P modelling): „BDE209 was not considered in the different modelling approaches for two main reasons. Firstly, higher uncertainties are associated with the measured particulate fractions for this compound (see Section 3.1). Secondly, two of the tested models are based on $K_{OA}$ and the temperature dependence of this parameter is not available (never determined). "*

Yang et al., 2018 provides KOA values and their temperature dependence for all 209 PBDE congeners. I would suggest the authors considering this recent publication and include BDE in their g/p partitioning modeling.

P.5. L. 27: the measured $f_{OM}$ value for this site shall be provided.

We have used $f_{OM}$ values provided by the Czech Hydrometeorological Institute which were measured every sixth day at the sampling site.

*The manuscript now includes: „The $f_{OM}$ were derived from the atmospheric concentrations of organic carbon (a conversion factor of 1.8 was used) which was determined every sixth day and were ranging from 0.07 to 0.98 with an average value of 0.39 ± 0.19."*

Summary statistics for the $f_{OM}$ are now provided. However it is not clear which value was used in the modelling of gas-particle partitioning, the average over the 4-year study or the corresponding weekly average?
Also, please check the value of 0.98, it seems to be very high even for PM from a background site.
Finally, please, correct the *"The $f_{OM}$ were derived from…"* to *"The $f_{OM}$ values were calculated from …"*.

P.7, L. 5: The average gas- and particle-phase concentrations of BDE209 provided in Table S5 (0.513 and 0.257 pg m$^{-3}$, respectively) seem to be in discrepancy with the average measured particulate fraction ($\theta_{measured}$) presented in Figure S6, which ranges between 55-85% in the four seasons. Please, check and correct if needed.

Indeed, these two datasets are in discrepancy, but correct. The average gaseous concentration of BDE209 was biased by few outliers (characterised by the high SD). The seasonal mean particulate mass fraction (FigureS3) was derived from the particulate mass fractions of individual samples. No changes made.

Please, expand axis Y of Figure S3 so as the full SD is shown. In my opinion, in data sets with very large SD the mean value is more representative than the average.

P.7, L. 9, 15, 16, 18: Besis and Samara, 2012 is not in the reference list. Actually, Besis and Samara 2012 is not dealing with the g/p partitioning of PBDEs. Perhaps the authors wanted to cite Besis et al., 2016 (Atmospheric occurrence and gas-particle partitioning of PBDEs at industrial, urban and suburban sites of Thessaloniki, northern Greece: Implications for human health, Envir. Poll. 215 (2016) 113-124).

Actually this section is not dealing with gas-particle partitioning and we consider that the information reviewed by Besis and Samara (2012) is relevant to support the points made with regard to the congener profiles. No changes made.

I am afraid that my comment was misunderstood. In this section, PBDE concentration levels found in this study are compared with those found in other European locations. Besis and Samara (2012) is a review article compiling literature data from all over the world and should be cited. I also suggested Besis et al. (2016) since it provides more recent data for European sites not included in the review.

Seasonality is confused here with the correlation with ambient T. Unfortunately, seasonal variations of PBDEs levels are not examined in the manuscript. Correlations with ambient T are as expected. Why the authors did not provide Clausius-Clapeyron plots for the gas-phase concentrations?

The investigation of seasonality on PBDEs atmospheric concentrations is now included in the Section 3.6.

I cannot see where the seasonal variations on PBDEs atmospheric concentrations are presented, either in the manuscript, or in the Supplementary Material. in Section 3.6, there is only a paragraph in P.13 *"The seasonal variations presented here are in contradiction with many previous studies which reported higher concentrations of most PBDEs in summer compared to winter……………………. enhance the revolatilisation from surfaces"*. Please, correct this deficiency.

P.9. L. 28: Again seasonality is confused with the correlation with ambient T. Please, correct properly.

Changed accordingly.

I cannot see where the seasonal variations on PBDEs atmospheric concentrations are presented, either in the manuscript, or in the Supplementary Material. in Section 3.6, there is only a paragraph in P.13 *"The seasonal variations presented here are in contradiction with many previous studies which reported higher concentrations of most PBDEs in summer compared to winter……………………. enhance the revolatilisation from surfaces"*. Please, correct this deficiency.

The authors could provide the logKp-T relationship as well in addition to the correlation coefficient between θmeasured and 1/T.

This is now included (Table S5).

The correct Table is S7. Please unbold the r2 value between θmeas and 1/T for BDE-209 as it not statistically significant.

**Additional comments on manuscript v.3**

**Abstract**

- Instead of providing only the average θmeasured for winter and summer for 2 PBDEs (without the corresponding SDs), I would suggest to provide a general description of the seasonal trend that for all PBDEs, except maybe BDE-209, seems to be winter > autumn ≈ spring > summer.
- Photolytic debromination was only assumed, not indicated by the results. Please, rephrase.

**Equation (5)**: Please, use capital letters for OA in $f_{om}$.

**P.5, L.22**: Please correct "The fOM were derived from…" to "The fOM values were calculated from…".

**P.6, L.10**: please delete extra parenthesis and commas.

**P.6, L.23-24**: The statement "*Given that Bidleman and Tysklind (2018) demonstrated that when less than 50% is found in the lower PUF plug, the collected gaseous mass fractions should be larger than 90%,…*" is is not correct. Bidleman and Tysklind (2018) predicted that the collection efficiency of the gas phase exceeds 90% when the PUF2/PUF1 ratio is <0.5 (this means <33% in the lower PUF). Please, correct this point properly.

**P.7, L.24-33**: The authors highlight here the large seasonal variations found for θmeasured providing related information in Figures 1 and S3. I think that this finding, which is also highlighted in the conclusions section, deserves better discussion.

- Firstly, Fig. 1, that shows measured particulate fraction (θmeasured) for 4 PBDEs only on individual dates, is not essential and should be deleted, or replaced by S3 (please, expand axis Y of Figure S3, so as the full seasonal SDs are shown)
- Moreover, the larger seasonal differences found here in comparison to other studies need explanation. Large seasonal differences in θmeas should be expected for large seasonal differences in ambient temperature. What is the seasonal difference of ambient temperature in this study and in the cited studies? For instance, Besis et al. (2016), found θmeas

~25% lower in summer in comparison to winter for a difference in temperature of about 10-15 °C.

**P.8, L.14-15**: It is stated that *"none of the three model approaches successfully predicted Kp or ϑ for all individual PBDEs considered"*. Did the authors examined the seasonality of the predictabilities of the models? As observed in Besis et al. (2016), Θmeasured data were closer to Θpredicted by the $K_{OA}$ model in summertime samples as compared to the wintertime samples.

**P.8, L.28-30**: This sentence is not accurate. Besis et al. (2017) found that the steady-state model, when performed at a background site, was superior to predict G/P partitioning of BDE-209, while the KOA model was comparable or slightly better than the steady-state model for BDE-66 and BDE-154.

**P.11, L.6**: The reverse correlation found in your study between Cp of several PBDE congeners and precipitation shall be commented here.

**P.11, L.23-25**: The suggestion *"Therefore, we would suggest to focus the interpretation of Clausius Clapeyron equation only for those substances which are mainly in the gas-phase (i.e. ϑmeasured < 0.2), regardless of the ambient temperature"* is unclear. In the present study, only BDE-28 has θmeasured < 0.2 regardless of the ambient T, but does not follow the C-C equation.

**P.12, L.8-9**: There is no any evidence in this study that could support the statement that combustion can be concluded as a primary PBDE emission source. The results indicated increased Cp concentrations for PBDEs in winter, but this does not necessarily mean emission from combustion sources. Please, correct this sentence.

**Conclusions**
- Please, keep only conclusions that are supported by your results.
- Please, note that the important finding of this study, i.e. the seasonal variation of the particulate fraction which was observed for most PBDEs, that is significantly larger than in other studies, was not adequately explained in the manuscript!
- The critique on passive sampling designs is pointless here. I suggest just highlighting the prevalent congeners in each phase that were found in this study.
- Please, give again the names of the *"available models"*.

**Supplementary Material**
Table S2: I am very confused about the LOQs as reported in this Table.
- I cannot understand the meaning of iLOQs expressed in pg/sample (suppose per filter or PUF plug) or in $pg/m^3$. These

should be referred as method LOQs. I would suggest the authors providing the iLOQ for each compound in pg/μL.

- What is the usefulness of calculating LOQblanks in pg/m$^3$ since field blanks are not subjected to air sampling?
- Furthermore, the LOQ of a specific measurand in field blanks includes the iLOQ, as a consequence it cannot be zero. Please, correct or clarify Table S2.

---

## Referee Report (RR2)

**Second review for manuscript titled "Are atmospheric PBDE levels declining in Central Europe? Examination of the seasonal variations, gas-particle partitioning and implications for long-range atmospheric transport" by Degrendele et al. (acp-2018-144)**

The authors have taken into account all reviewers' questions and comments and have significantly improved the manuscript by providing more information regarding blanks and QA/QC procedures and corrected errors. I think the manuscript can be accepted for publication after the following minor issues are addressed:

P. 1, Line 20, line 28 and elsewhere in the text: Suggest that whenever the authors refer to the PBDE congeners, list the congener numbers from lowest to the highest. It is very strange to give "…found for BDE 100, 99, 153 and 209", why not "…found for BDE 99, 100, 153 and 209"? Also, line 28, "…BDE99 and BDE 47", why not "BDE 47 and 99"? It makes it easier to read.

p. 5 line 22 where did the conversion factor of 1.8 come from? What unit were the atmospheric concentrations of organic carbon given in?

p. 6 I presume that the vapor pressures given are at 25 °C. Please state this in the text.

Table S8 Should change to Spearman Correlation results.

p. 11, line 9 Please specify that the ABL is given as hmix in Table S9.

p. 12, last line "Atmospheric boundary layer" was the term used earlier. Better use the same term to be consistent.

P. 20 Figure 1 The blue bars are too light and difficult to read. Please make them darker.

p. 23 Figure 4 I am wondering why the authors did not choose to show BDE 100 and 153, which show statistically significant trends, in this figure instead of BDE 47 and 183 that have no statistically significant trends.

Minor:
P. 1, Line 29 Usually "accounting for approximately…"
P. 3 Line 4 "gas-phase chemicals on …"
p. 3 Line 8 "…discarded without further analysis…"
p. 5 line 27 spell out above ground level (a.g.l.)
p. 5 line 3 and elsewhere : Li et al. <no comma> (2015)
p. 7 last line: should read "…LRAT potential of PBDEs and for developing…"
p. 10, line 11 …the precipitation rate were found for all PBDEs…
p. 11, line 22 missing space between "gaseous" and "concentrations"

p. 11, line 26 "On the other hand,…"
Table S3 Sample preparation, not Sampling preparation.

---

## Author Response (AR2)

Once more, we would like to sincerely thank the reviewers for their precious comments, which, again, improved this manuscript. We have addressed every point below and have indicated the corresponding modifications in a revised version of the manuscript.

**Second review for manuscript titled "Are atmospheric PBDE levels declining in Central Europe? Examination of the seasonal variations, gas-particle partitioning and implications for long-range atmospheric transport" by Degrendele et al. (acp-2018-144)**
The authors have taken into account all reviewers' questions and comments and have significantly improved the manuscript by providing more information regarding blanks and QA/QC procedures and corrected errors. I think the manuscript can be accepted for publication after the following minor issues are addressed:

P. 1, Line 20, line 28 and elsewhere in the text: Suggest that whenever the authors refer to the PBDE congeners, list the congener numbers from lowest to the highest. It is very strange to give "…found for BDE 100, 99, 153 and 209", why not "…found for BDE 99, 100, 153 and 209"? Also, line 28, "…BDE99 and BDE 47", why not "BDE 47 and 99"? It makes it easier to read.
Thanks, now corrected accordingly.

p. 5 line 22 where did the conversion factor of 1.8 come from? What unit were the atmospheric concentrations of organic carbon given in?
Usually, a conversion factor of 1.4 has been commonly used in aerosol studies. However, we have decided to use 1.8, based on a previous study measuring OC and OM at different US national parks in which the authors suggested that the value of 1.4 was too low for atmosheric research.
The manuscript now reads: „The $f_{OM}$ were calculated from the atmospheric concentrations of organic carbon (available in $\mu g \ m^{-3}$, a conversion factor from organic carbon to OM of 1.8 was used, El-Zanan et al., 2005) which were determined every sixth day. The corresponding weekly averages were used and ranged from 0.07 to 0.98 with an average value of $0.39 \pm 0.19$.."

p. 6 I presume that the vapor pressures given are at 25 °C. Please state this in the text.
Done accordingly.

Table S8 Should change to Spearman Correlation results.
Done accordingly.

p. 11, line 9 Please specify that the ABL is given as hmix in Table S9.
Thanks, we have now replaced hmix by ABL height in Table S9.

p. 12, last line "Atmospheric boundary layer" was the term used earlier. Better use the same term to be consistent.
Thanks, now corrected accordingly.

P. 20 Figure 1 The blue bars are too light and difficult to read. Please make them darker.
Thanks, now corrected accordingly.

p. 23 Figure 4 I am wondering why the authors did not choose to show BDE 100 and 153, which show statistically significant trends, in this figure instead of BDE 47 and 183 that have no statistically significant trends.

In this Figure 4 as well as in Figure 1, we have chosen to focus on the compounds showing the highest concentrations (i.e. > 80% of $\Sigma_9$PBDEs), while the remaining BDEs are presented in Figure S11. Presenting only compounds that show a significant decrease in 2011-2014 in the manuscript could be misleading for the readers.

Minor:
P. 1, Line 29 Usually "accounting for approximately…"
Done accodingly.
P. 3 Line 4 "gas-phase chemicals on …"
Done accodingly.
p. 3 Line 8 "…discarded without further analysis…"
Done accodingly.
p. 5 line 27 spell out above ground level (a.g.l.)
Done accodingly.
p. 5 line 3 and elsewhere : Li et al. <no comma> (2015)
Done accodingly.
p. 7 last line: should read "…LRAT potential of PBDEs and for developing…"
Done accodingly.
p. 10, line 11 …the precipitation rate were found for all PBDEs…
Done accodingly.
p. 11, line 22 missing space between "gaseous" and "concentrations"
Done accodingly.
p. 11, line 26 "On the other hand,…"
Done accodingly.
Table S3 Sample preparation, not Sampling preparation.
Changed accodingly.

MS No.: acp-2018-144
Title: Are atmospheric PBDE levels declining in Central Europe? Examination of the seasonal variations, gas-particle partitioning and implications for long-range atmospheric transport
Author(s): Céline Degrendele et al.
MS Type: Research article
General comments on the authors' response and the revised manuscript
The revised manuscript is substantially improved, however needs further perfections before it can be accepted for publication in acp.
The authors' response (in green) was reasonable to most of my comments, although some issues remain unsolved (see below in red). In addition, there are several other issues raised that need to be addressed (see Additional comments on manuscript v.3).

Comments on the original manuscript needing further attention
Another question is why the subcooled-liquid–vapor pressure (PL)-based model was excluded from the g/p partitioning analysis.
We refrained from exploring log Kp = f(log pL) as the temperature dependence of vapour pressure is also reflected in the log Kp = f(log Koa) plots (see e.g. Pankow and Bidleman, 1992; Cetin and Odabasi 2008; Lammel et al., 2010). Previously, it was common to test another vapour pressure based model i.e., the Junge-Pankow adsorption model (Pankow 1987). Such a model, implicitly assuming that adsorption is dominating gas-particle partitioning of the substances under study, is generally not promising for hydrophobic substances, which gas-particle partitioning is expected to be dominated by absorption in particulate organic matter

(Finizio et al., 1997; Lohmann and Lammel, 2004; Goss and Schwarzenbach, 2001). The Junge-Pankow model has nevertheless been tested for PBDEs (Chen et al., 2006) including on another set of aerosol samples we collected and analysed (Besis et al., 2017). These results had confirmed the deficiency of this model and the perception that adsorption is not a significant process for PBDE gas-particle partitioning. Therefore, we prefer to not include this model in the discussion on gasparticle partitioning.

I agree that adsorption is less significant that absorption, nevertheless, the Junge-Pankow adsorption model was found to predict better than the KOA model the θmeasured in the warm season for the moderately brominated congeners DBE-49, -71, -47, and -66 (Besis et al., 2016). We now have applied the Junge-Pankow model as suggested (input data. $c_J$ = 17.2 Pa cm and aerosol surface S = 5 x $10^{-5}$ $cm^2$ $cm^{-3}$, measured at the site; Shahpoury et al; 2016 ES&T). The model overpredicts partitioning largely (particulate mass fractions 0.70 (0.33-0.94) for BDE28 and > 0.9 for all other congeners (Figure 1 below).

[Figure]

Figure 1: Measured and predicted particulate fraction of all PBDEs investigated for the average conditions at the sampling site.

This result was anticipated, and is similar to previous results (Besis et al., 2017). Such results just confirm the understanding achieved in recent years, that adsorption is less significant than absorption for PBDEs (and other lipophilic semivolatiles), The advancement of understanding gas-particle partitioning is now better phrased in the text (section 2.4).

The manuscript now reads: „$K_{OA}$ model (...) assumes that (...) absorption into particulate organic matter (OM) of the particles determines the distribution process, while other types of molecular interaction (i.e. adsorption to the unspecific surface, to minerals or soot) are negligible (Harner and Bidleman, 1998a).

The exclusion of BDE209 from all g/p partitioning models needs explanation.

Two of the presented models used KOA as one of the critical parameter. To the best of our knowledge, given the analytical issues with BDE209, there are no measured KOA as a function of temperature for this compound available. For all remaining BDEs, we have used measured KOA relationships. It is therefore evident that an estimation of KOA as a function of the temperature will be associated with higher uncertainties than the measured values. Moreover, there are higher uncertainties with the reported measured particulate fraction for BDE209, we therefore prefered to exclude this compound from the G/P modelling.

The manuscript now includes (at the beginning of the section on G/P modelling): „BDE209 was not considered in the different modelling approaches for two main reasons. Firstly, higher

uncertainties are associated with the measured particulate fractions for this compound (see Section 3.1). Secondly, two of the tested models are based on KOA and the temperature dependence of this parameter is not available (never determined). "

Yang et al., 2018 provides KOA values and their temperature dependence for all 209 PBDE congeners. I would suggest the authors considering this recent publication and include BDE in their g/p partitioning modeling.

Thank you very much for indicating this data source. We followed the suggestion and used the $K_{OA}(T)$ estimates of Yang et al., 2018, to predict gas-particle partitioning according to the different models investigated also for BDE209 (Figure 2, below). A large discrepancy to observed values, systematic overestimate of the particulate mass fraction is found for the $K_{OA}$-model.

[Figure]

*Figure 2: Comparison of measured and predicted logKp of BDE209. Theblue line reprersents the 1:1 line*

For congeners with experimental $K_{OA}(T)$ data available (Shoeib and Harner, 2002), we tested on the implied uncertainties of estimated $K_{OA}(T)$. For the highest congener most similar to BDE209, i.e. BDE183, an octaBDE, discrepancies of up to two orders of magnitude are suggested (Figure 3, below). The same deviation from the true $K_{OA}$ value can be expected for BDE209.

[Figure]

*Figure 3: Comparison of $logK_{OA}$ predicted (estimates, (Yang et al., 2018) and measured (exp., Shoeib and Harner, 2002). The results shown are for a temperature range of 253-303 K.*

Moreover, the data set of observed BDE209 concentrations in both phases is a particularly small one (32% of the measurements) and is associated with large uncertainties (discussed in section 3.1), and was therefore excluded from this part of the study. Considering this and the large uncertainty of the model input data ($K_{OA}(T)$; Fig. 3), we still are not sufficiently confident to include these results in the current article.

P.5. L. 27: the measured fOM value for this site shall be provided.
We have used fOM values provided by the Czech Hydrometeorological Institute which were measured every sixth day at the sampling site. The manuscript now includes: „The fOM were derived from the atmospheric concentrations of organic carbon (a conversion factor of 1.8 was used) which was determined every sixth day and were ranging from 0.07 to 0.98 with an average value of $0.39 \pm 0.19$.“
Summary statistics for the fOM are now provided. However it is not clear which value was used in the modelling of gas-particle partitioning, the average over the 4-year study or the corresponding weekly average?
Thanks. Followed.
The manuscript now reads: „The corresponding weekly averages were used and ranged from 0.07 to 0.98 with an average value of $0.39 \pm 0.19$.“

Also, please check the value of 0.98, it seems to be very high even for PM from a background site.
We recognize that this value is extremely high for a background site, and this may be due to our choice for the conversion factor of 1.8 from organic carbon to organic matter. For this weekly sample, the $f_{OC}$ was 0.545 (i.e. OC concentration was 3.27 µg/m$^3$ and PM$_{10}$ concentration was 6 µg/m$^3$). No changes were done.

Finally, please, correct the "The fOM were derived from…" to "The fOM values were calculated from …".
Done accordingly.

P.7, L. 5: The average gas- and particle-phase concentrations of BDE209 provided in Table S5 (0.513 and 0.257 pg m-3, respectively) seem to be in discrepancy with the average measured particulate fraction (θmeasured) presented in Figure S6, which ranges between 55-85% in the four seasons. Please, check and correct if needed.

Indeed, these two datasets are in discrepancy, but correct. The average gaseous concentration of BDE209 was biased by few outliers (characterised by the high SD). The seasonal mean particulate mass fraction (FigureS3) was derived from the particulate mass fractions of individual samples. No changes made.

Please, expand axis Y of Figure S3 so as the full SD is shown. In my opinion, in data sets with very large SD the mean value is more representative than the average.

Done accordingly. We have now expanded the Y-axis to 1.2, such that we can fully see the SD.

P.7, L. 9, 15, 16, 18: Besis and Samara, 2012 is not in the reference list. Actually, Besis and Samara 2012 is not dealing with the g/p partitioning of PBDEs. Perhaps the authors wanted to cite Besis et al., 2016 (Atmospheric occurrence and gasparticle partitioning of PBDEs at industrial, urban and suburban sites of Thessaloniki, northern Greece: Implications for human health, Envir. Poll. 215 (2016) 113-124).

Actually this section is not dealing with gas-particle partitioning and we consider that the information reviewed by Besis and Samara (2012) is relevant to support the points made with regard to the congener profiles. No changes made.

I am afraid that my comment was misunderstood. In this section, PBDE concentration levels found in this study are compared with those found in other European locations. Besis and Samara (2012) is a review article compiling literature data from all over the world and should be cited. I also suggested Besis et al. (2016) since it provides more recent data for European sites not included in the review.

Thanks for clarifying, and sorry for the misunderstanding. We have now added these two references.

Seasonality is confused here with the correlation with ambient T. Unfortunately, seasonal variations of PBDEs levels are not examined in the manuscript. Correlations with ambient T are as expected. Why the authors did not provide Clausius-Clapeyron plots for the gas-phase concentrations?

The investigation of seasonality on PBDEs atmospheric concentrations is now included in the Section 3.6.

I cannot see where the seasonal variations on PBDEs atmospheric concentrations are presented, either in the manuscript, or in the Supplementary Material. in Section 3.6, there is only a paragraph in P.13 "The seasonal variations presented here are in contradiction with many previous studies which reported higher concentrations of most PBDEs in summer compared to winter……………………………. enhance the revolatilisation from surfaces". Please, correct this deficiency.

This is a misunderstanding. As suggested by reviewer #3, we have now used the opportunity of analyzing the semi-long term trends along with the seasonality using the regression model proposed by Venier et al; (2012). The paragraph preceeding the one mentioned here was presenting these seasonal differences which were statistically significant using the regression model (paragraph in p.13: „Unlike a recent study … ABL height is dominating". No changes were done.

P.9. L. 28: Again seasonality is confused with the correlation with ambient T. Please, correct properly.

Changed accordingly.

I cannot see where the seasonal variations on PBDEs atmospheric concentrations are presented, either in the manuscript, or in the Supplementary Material. in Section 3.6, there is only a paragraph in P.13 "The seasonal variations presented here are in contradiction with many previous studies which reported higher concentrations of most PBDEs in summer compared to winter……………………………. enhance the revolatilisation from surfaces". Please, correct this deficiency.

This is a misunderstanding. This comment was related to the seasonality of the measured particulate fraction, which has already been improved in the first revision incorporating the seasonal variations (Figure S3 and related discussion in the manuscript). No changes were done.

The authors could provide the logKp-T relationship as well in addition to the correlation coefficient between $\theta$measured and 1/T.

This is now included (Table S5).

The correct Table is S7. Please unbold the $r^2$ value between $\theta$meas and 1/T for BDE-209 as it not statistically significant.

We apologize for these errors. Table S7 has been changed accordingly.

Additional comments on manuscript v.3

Abstract

☐ Instead of providing only the average $\theta$measured for winter and summer for 2 PBDEs (without the corresponding SDs), I would suggest to provide a general description of the seasonal trend that for all PBDEs, except maybe BDE-209, seems to be winter > autumn ≈ spring > summer.

Followed. We have now added a more general description of the seasonal trend of the gas-particle partitioning as well as the standard deviations of the reported particulate fractions for BDE47 and BDE99.

The abstract now reads: „Clear seasonal variations with significantly higher measured particulate fraction ($\theta_{measured}$) in winter compared to summer was found for all PBDEs except BDE209. For example, while the average $\theta_{measured}$ of BDE47 was $0.53 \pm 0.19$ in winter, this was only $0.01 \pm 0.02$ in summer. Similarly, for BDE99, $\theta_{measured}$ was $0.89 \pm 0.13$ in winter, while it was only $0.12 \pm 0.08$ in summer."

☐ Photolytic debromination was only assumed, not indicated by the results. Please, rephrase.

Followed.

The abstract now reads: „The results suggest that photolytic debromination … atmosphere"

Equation (5): Please, use capital letters for OA in fom.

Changed accordingly.

P.5, L.22: Please correct "The fOM were derived from…" to "The fOM values were calculated from…".

Changed accordingly.

P.6, L.10: please delete extra parenthesis and commas.

Changed accordingly.

P.6, L.23-24: The statement "Given that Bidleman and Tysklind (2018) demonstrated that when less than 50% is found in the lower PUF plug, the collected gaseous mass fractions should be larger than 90%,…" is is not correct. Bidleman and Tysklind (2018) predicted that the collection efficiency of the gas phase exceeds 90% when the PUF2/PUF1 ratio is <0.5 (this means <33% in the lower PUF). Please, correct this point properly.

We apologize for this error. This has been changed accordingly.

P.7, L.24-33: The authors highlight here the large seasonal variations found for θmeasured providing related information in Figures 1 and S3. I think that this finding, which is also highlighted in the conclusions section, deserves better discussion.

☐ Firstly, Fig. 1, that shows measured particulate fraction (θmeasured) for 4 PBDEs only on individual dates, is not essential and should be deleted, or replaced by S3 (please, expand axis Y of Figure S3, so as the full seasonal SDs are shown)

This is a misunderstanding. Figure 1 shows the complete time series of the experimentally determined particulate mass fraction of the four most abundant PBDE congeners. This is the only figure which shows the time series, while Figure S3 shows seasonally aggregated data. For clarification, we improved the layout. Y-axis expanded in Figure S3, as suggested.

☐ Moreover, the larger seasonal differences found here in comparison to other studies need explanation. Large seasonal differences in θmeas should be expected for large seasonal differences in ambient temperature. What is the seasonal difference of ambient temperature in this study and in the cited studies? For instance, Besis et al. (2016), found θmeas ~25% lower in summer in comparison to winter for a difference in temperature of about 10-15 oC.

The large seasonal differences of the measured particulate fraction of PBDEs found in this study (T is ranging from -6.4 to 23.0 °C for individual samples) are similar to the one reported at the Great Lakes with a similar temperature range (from -9.9 to 23 °C), but higher than those measured in Greece (difference in temperature of 8-13 °C) or in the Arctic (difference in temperature of 18°C).

The manuscript now reads: „The large seasonal differences in the gas-particle partitioning of PBDEs have been previously reported from a rural site in the American Great Lakes area where the differences in the ambient temperatures were similar to this study, about 30 °C (Su et al., 2009). Less seasonal difference in gas-particle partitioning was found in the Arctic or in Greece where the temperature range was lower than 20 °C (Besis et al., 2016; Davie-Martin et al., 2016)"

The following sentence, which conludes on the relevance of the temperature dependence of gas-particle partitioning we removed, as this emphasis is not needed here.

P.8, L.14-15: It is stated that "none of the three model approaches successfully predicted Kp or θ for all individual PBDEs considered". Did the authors examined the seasonality of the predictabilities of the models? As observed in Besis et al. (2016), Θmeasured data were closer to Θpredicted by the KOA model in summertime samples as compared to the wintertime samples.

We also evaluated the efficiency of these models for the summer and winter sample subsets (new Figure S5), and it is evident that none of these models successfully captured the particulate fraction for all PBDEs in a specific season (in Figure S5, the individual dots are not scattered around the 1:1 line).

The manuscript now reads: „As presented in Figures 2 and S4, none of the three model approaches successfully predicted $K_p$ or θ for all individual PBDEs considered, which is also the case when considering only winter or summer samples (Figure S5)."

P.8, L.28-30: This sentence is not accurate. Besis et al. (2017) found that the steady-state model, when performed at a background site, was superior to predict G/P partitioning of BDE-209, while the KOA model was comparable or slightly better than the steady-state model for BDE-66 and BDE-154.

Sorry about this mistake. We have now changed accordingly the text.

The manuscript now reads: „The only other study test of this model on atmospheric PBDE data did not find an acceptable performance for all PBDEs investigated, although, for BDE209 it predicted better than the $K_{OA}$-model (Besis et al., 2017)"

P.11, L.6: The reverse correlation found in your study between Cp of several PBDE congeners and precipitation shall be commented here.

We are not convinced that adding the correlation between Cp of individual PBDEs and precipitation rate (presented in Section 3.3) also in this section would add clarity, but rather would disturb the reader from the original message (i.e. predicting $K_p$ within one order of magnitude is not sufficiently accurate for characterizing phase-specific removal processes). However, we have now provided an example of this phase-specific removal process (i.e. wet scavenging of particles) and made a reference to the previous section.

The manuscript now reads: „Therefore, these models are not ideal when phase-specific removal processes such as the wet scavenging of particles (see Section 3.3) are to be estimated."

P.11, L.23-25: The suggestion "Therefore, we would suggest to focus the interpretation of Clausius Clapeyron equation only for those substances which are mainly in the gas-phase (i.e. θmeasured < 0.2), regardless of the ambient temperature" is unclear. In the present study, only BDE-28 has θmeasured < 0.2 regardless of the ambient T, but does not follow the C-C equation.

This is exactly our point. If we would consider all PBDEs, the results would suggest that BDE28 and BDE209 are influenced by LRAT while the other congeners are influenced by revolatilisation from the surfaces. However, given the important seasonal variations of $\theta_{measured}$ for all PBDEs except BDE28 and BDE209, it is not clear whether the influence of ambient temperature on the gaseous concentrations are due to air-surface exchange rather than revolatilisation from the particles. Therefore, the only valid conclusion that can be taken from this study is that the gaseous BDE28 concentrations were controled by LRAT rather than by air surface exchange.

The manuscript now reads: „Following this, we can only conclude from the present study that the gaseous concentrations of BDE28 were not controlled by air-surface exchange."

P.12, L.8-9: There is no any evidence in this study that could support the statement that combustion can be concluded as a primary PBDE emission source. The results indicated increased Cp concentrations for PBDEs in winter, but this does not necessarily mean emission from combustion sources. Please, correct this sentence.

This has been changed accordingly.

The manuscript now reads: „In conclusion, the atmospheric concentrations of individual PBDEs were controlled by deposition processes (wet scavenging), meteorological parameters (ABL height, temperature) and LRAT while the influence of revolatilisation could not be demonstrated."

Conclusions
☐ Please, keep only conclusions that are supported by your results.

Followed: We have now removed the sentence that atmospheric PBDE levels were governed by primary emissions, which was not supported by our results.

☐ Please, note that the important finding of this study, i.e. the seasonal variation of the particulate fraction which was observed for most PBDEs, that is significantly larger than in other studies, was not adequately explained in the manuscript!

Followed: Now more emphasised in the discussion of the main results part of the manuscript, see Section 3.3 and previous comment.

☐ The critique on passive sampling designs is pointless here. I suggest just highlighting the prevalent congeners in each phase that were found in this study.

Misunderstanding: The critique was not on the passive sampling design, but rather on the interpretation of results for semi-volatile organic compounds' levels obtained from such passive devices. In fact, many studies interpreted the summer maxima organics concentrations based

on passive techniques as the proof that their atmospheric concentrations are dominated by air-surface exchange, regardless of the shift in gas-particle partitioning.

The manuscript now reads: „Therefore, the interpretation of the seasonal variations of PBDEs from such studies should be done in a cautious manner, distinguishing whether the increased concentrations are due to gas-particle partitioning shift or to increased secondary emissions."

☐ Please, give again the names of the "available models".

Changed accordingly.

Supplementary Material

Table S2: I am very confused about the LOQs as reported in this Table.

☐ I cannot understand the meaning of iLOQs expressed in pg/sample (suppose per filter or PUF plug) or in pg/m3. These should be referred as method LOQs. I would suggest the authors providing the iLOQ for each compound in pg/μL.

Followed: We have now provided all iLOQs in pg/ μL.

☐ What is the usefulness of calculating LOQblanks in pg/m3 since field blanks are not subjected to air sampling?

Followed: $LOQ_{blanks}$ now presented n pg/ μL throughout.

☐ Furthermore, the LOQ of a specific measurand in field blanks includes the iLOQ, as a consequence it cannot be zero. Please, correct or clarify Table S2.

Sure. LOQs were indeed determined as the maximum between iLOQs and $LOQ_{blanks}$. For clarification, we have now replaced the 0 values for $LOQ_{blanks}$ by ND.